# Proanthocyanidins: Impact on Gut Microbiota and Intestinal Action Mechanisms in the Prevention and Treatment of Metabolic Syndrome

**DOI:** 10.3390/ijms24065369

**Published:** 2023-03-10

**Authors:** Rocío Redondo-Castillejo, Alba Garcimartín, Marina Hernández-Martín, María Elvira López-Oliva, Aránzazu Bocanegra, Adrián Macho-González, Sara Bastida, Juana Benedí, Francisco J. Sánchez-Muniz

**Affiliations:** 1Pharmacology, Pharmacognosy and Botany Department, Pharmacy School, Complutense University of Madrid, 28040 Madrid, Spain; 2Departmental Section of Physiology, Pharmacy School, Complutense University of Madrid, 28040 Madrid, Spain; 3Nutrition and Food Science Department (Nutrition), Pharmacy School, Complutense University of Madrid, 28040 Madrid, Spain

**Keywords:** metabolic syndrome management, proanthocyanidins, bioactive compounds, dysbiosis, prevention strategy, treatment strategy, intestinal barrier integrity

## Abstract

The metabolic syndrome (MS) is a cluster of risk factors, such as central obesity, hyperglycemia, dyslipidemia, and arterial hypertension, which increase the probability of causing premature mortality. The consumption of high-fat diets (HFD), normally referred to high-saturated fat diets, is a major driver of the rising incidence of MS. In fact, the altered interplay between HFD, microbiome, and the intestinal barrier is being considered as a possible origin of MS. Consumption of proanthocyanidins (PAs) has a beneficial effect against the metabolic disturbances in MS. However, there are no conclusive results in the literature about the efficacy of PAs in improving MS. This review allows a comprehensive validation of the diverse effects of the PAs on the intestinal dysfunction in HFD-induced MS, differentiating between preventive and therapeutic actions. Special emphasis is placed on the impact of PAs on the gut microbiota, providing a system to facilitate comparison between the studies. PAs can modulate the microbiome toward a healthy profile and strength barrier integrity. Nevertheless, to date, published clinical trials to verify preclinical findings are scarce. Finally, the preventive consumption of PAs in MS-associated dysbiosis and intestinal dysfunction induced by HFD seems more successful than the treatment strategy.

## 1. Metabolic Syndrome: General Aspects

Metabolic syndrome (MS) is a condition of chronic low-grade inflammation, including a cluster of interrelated metabolic factors like insulin resistance, hyperglycemia, atherogenic dyslipidemia, central obesity, arterial hypertension, and proinflammatory and/or prothrombotic state [1]. The increased incidence of these metabolic factors is a huge public health problem. An untreated and persistent MS state can lead to several pathologies, such as type 2 diabetes mellitus (T2DM), atherosclerotic cardiovascular disease, stroke, and kidney disease, among others [2].

MS is considerably prevalent and widely distributed, both in developing and developed countries. This can be attributed in part to several modifiable risk factors, such as obesity, sedentary lifestyle, excess energy intake (saturated fat, trans fatty acids, and sugary drinks), alcohol, smoking, and some environmental factors such as stress. Furthermore, there are other non-changeable risk factors, such as family history, genetic predisposition, age, or postmenopausal status, among others [2,3]. Therefore, prevention and early diagnosis are crucial to reduce risk factors and change lifestyles. Although there is no consensus, it is widely accepted that treatment of MS should be multifactorial, including dietary guidelines and physical activity in all cases, which can be accompanied by pharmacological treatment [2].

The definition of MS and diagnostic criteria have changed over time. In recent years, different organizations have recommended diagnostic criteria that apply basic clinical and analytical parameters for the diagnosis of MS. The most widely used criteria are those of the World Health Organization (WHO), the National Cholesterol Education Program (NCEP), the European Group for the study of Insulin Resistance (EGIR), the American Association of Clinical Endocrinologist (AACE), and the International Diabetes Federation (IDF). To end the coexistence of different criteria with overlapping parameters, in 2009, a group of organizations proposed unifying the diagnosis criteria of MS [1]. They created the harmonizing criterion, in which the presence of the factors is not mandatory and the cut-off point for them all was revised. The harmonizing criterion for MS diagnosis comprises five parameters, and to accept that a patient has MS must exceed the defined cut-offs in at least three of them. (1) A large waist circumference; used as an indicator of central obesity and the cut-off point is population, ethnicity, and the gender specificity. (2) A high plasma triglyceride level ≥150 mg/dL or receiving hypolipidemic drug treatment. (3) A low plasma HDL-cholesterol level with <40 mg/dL in men and <50 mg/dL in women, or receiving hypocholesterolemic drug treatment. (4) High blood pressure with ≥130 mmHg of systolic pressure or ≥85 mmHg of diastolic pressure, or receiving antihypertensive drug treatment. (5) High fasting blood glucose with ≥100 mg/dL, or receiving antidiabetic drug treatment. Furthermore, these health organizations specify that, according to the proposed criterion, most patients with T2DM may also have MS. Moreover, data from our group suggest that most patients with diabetes also have MS because besides hyperglycemia, low HDL-cholesterol, high blood triglycerides, and central obesity are usually present [4].

## 2. Loss of Intestinal Homeostasis as an Etiological Factor of Metabolic Syndrome and the Importance of Consuming a High-Fat Diet

Interactions between diet (macronutrients and micronutrients), gut microbiota (GM), and mucosal barriers have been reported to be of great importance in the onset and/or progression of major immune/metabolic disorders. So much so, it had been traditionally accepted that adipose tissue was mainly responsible for the inflammatory state and metabolic alterations of MS, but recent evidence indicates that gut barrier disturbances could be the origin of MS pathologies [5]. In this regard, the disruption of the intestinal barrier leads to translocation of microbes and endotoxins into the mucosa, which triggers inflammation and immune responses and can promote systemic metabolic diseases [6,7,8,9].

Following an unbalanced diet, such as high-fat diets (HFD), which normally referred to high-saturated fat diets, increases the risk of developing MS, in part because it alters GI homeostasis. An increased intake of total energy or saturated fat is considered one of the main drivers of the increased incidence of metabolic diseases. Therefore, the following is a brief description of the alterations promoted by HFD consumption in (a) the composition of GM and the changes in the metabolites it produces (especially SCFAs), (b) the integrity of the intestinal barrier, and (c) endotoxemia. Figure 1 indicates the main alterations at the intestinal level following the HFD consumption involved in the development of MS (Figure 1).

### 2.1. Intestinal Dysbiosis and Alterations in the SCFAs Profile Promoted by High-Fat Diets and Associated with Metabolic Syndrome

The Western diet has been associated with intestinal dysbiosis and an altered short-chain fatty acid (SCFA) profile, leading to loss of GM host homeostasis. HFD consumption causes changes in the diversity of dominant gut bacteria and modifies the bacterial profile toward less beneficial species, contributing to intestinal inflammation and the development of MS [10]. Many animal models and clinical trials of MS, although with controversial conclusions, have shown increases in the number of strains belonging to the phyla *Firmicutes* and *Proteobacteria*, related to the inflammatory response, at the expense of a reduction in the number of species of the phylum *Bacteroidetes* [5,10,11,12]. Therefore, an increased *Firmicutes*/*Bacteroidetes* (F/B) ratio is an accepted marker of obesity-associated dysbiosis [13]. This dysbiosis is considered to play a key role in the origin and development of MS, representing the main environmental factor [14].

However, it must be considered that GM produces an array of metabolites through the breakdown of nondigestible carbohydrates and some proteins that contribute to energy harvest. The most abundant of these metabolites are SCFAs: acetate, propionate, and butyrate. The profile and concentration of SCFAs generated depend on the composition of the diet and the GM. Therefore, HFD-induced dysbiosis, such as the Western diet, leads to an altered SCFA profile, which has also been linked to metabolic diseases such as obesity and MS [15].

SCFAs directly modulate host health through a variety of tissue-specific mechanisms related to gut barrier function, glucose homeostasis, immunomodulation, and appetite regulation. Their important role in systemic functions explains why SCFAs treatment, composed mainly of butyrate and propionate, has been shown to improve insulin sensitivity and glucose homeostasis in animal models of MS [16]. Increased circulating levels of propionate and butyrate have been linked to the regulation of the immune system and metabolism, the inhibition of tumor cell proliferation, and anti-inflammatory effects [16,17,18,19].

In contrast, recent evidence suggests that in MS and related diseases, such as obesity, the total plasma SCFA concentration is notably higher, with a particular pronounced increase in acetate [20,21,22]. Some studies show that the diet-related changes of SCFAs are highly complex and controversial, and their regulatory effects are still unknown [23,24]. It should be considered whether increased SCFA levels in blood could be related to beneficial or deleterious responses depending on their source of production. In this way, supplementation with dietary fibers, considering an appropriate strategy to promote intestinal health in T2DM, obesity, or MS, causes an increase in SCFA production [25] and their circulating levels [18,24] because of their prebiotic effect. In contrast, other researchers highlight that HFD consumption increases total SCFAs, mainly because of the rise in acetate, and links it with non-positive responses. From this viewpoint, the effects of SCFAs promote an aggravation of MS-related disorders [20]. Therefore, although acetate in in vitro and in vivo tests has a protective effect on the intestinal barrier, in MS animal models fed HFD, the chronic increase in plasma acetate contributes to a hypertriglyceridemic and hyperinsulinemic response, as well as insulin resistance, hyperphagia, and weight gain.

### 2.2. Colon Barrier Integrity Disruption Promoted by High-Fat Diets and Associated with Metabolic Syndrome

The intestinal barrier is a physical structure provided by different elements, both cellular and extracellular, which in coordination protects against harmful substances. Some of these elements are the commensal GM; the physical, and secretory barrier; the innate and adaptive immune system; and the nervous system, divided into enteric and central [26]. The physical barrier comprises epithelial cells joined by inter-epithelial tight junctions (TJs). Furthermore, the secretory barrier includes the mucus layer, antimicrobial peptides, and secretory immunoglobulin A. This complex network is critical to maintaining intestinal immune homeostasis.

Dysbiosis promoted by HFD consumption and associated with MS causes loss of intestinal barrier integrity, inducing the disruption of TJs and an increase in its permeability [27,28]. In this context, cells of gastrointestinal innate immunity (involving epithelial cells and antigen-presenting cells found in the lamina propria) that exhibit Toll-type receptors (TLRs) on their membrane recognize lipopolysaccharide (LPS) from Gram-negative bacteria and activate inflammatory pathways—primarily mitogen-activated protein kinases (MAPK) and nuclear factor kappa-light-chain-enhancer of activated B cells (NF-κB) pathways. This event leads to the overexpression of TLRs, the release of inflammatory mediators, and the transcription of inducible enzymes related with the inflammatory and oxidative process. Moreover, MAPK activation favors the blockage in the MLCK transcription, finally promoting the disorganization of the TJs and amplifying the pathological barrier hyperpermeability. In this sense, it has been demonstrated that proinflammatory mediators directly induce the TJ disruption, which dramatically increases intestinal permeability and facilitates bacterial translocation [9]. Oxidative stress and inflammatory cytokines overactivate the inflammatory pathways in a complex feedback loop, which act by amplifying barrier dysfunction and perpetuating the inflammatory state [26,28,29]. Consequently, dysbiosis and intestinal barrier dysfunction are one of the most important factors connecting diet and intestinal dysbiosis to the development of systemic metabolic disorders [9], including neuroinflammatory manifestations through the microbiota-gut-brain axis [30,31].

### 2.3. Metabolic Endotoxemia Promoted by High-Fat-Diets and Associated with Metabolic Syndrome

The adhesion of bacteria to the altered intestinal mucosa and its subsequent translocation, with the leakage of bacterial endotoxins such as LPS, has been identified as ultimately responsible for the inflammatory response and metabolic alterations observed in MS [7,9]. Bacteria can reach any peripheral tissue from the systemic circulation, so they have been reported to be detected, for example, in adipose tissue, resulting in the generalized inflammation that occurs in MS. This endotoxemia derived from HFD consumption is called “metabolic endotoxemia” and contributes to MS with a high metabolic impact. It favors bacterial translocation in a positive feedback loop, causing systemic disturbances [10,28,32]. Hence, intestinal dysbiosis and barrier dysfunction ultimately promote insulin resistance, obesity, inflammation, dyslipidemia, and even hypertension, all of them important features of MS [33].

## 3. Metabolic Syndrome Management

MS comprises some comorbidities that are better to treat in order to avoid further health consequences such as T2DM and cardiovascular disease. Changes in lifestyles and pharmacological treatment are the two main strategies contemplated in the MS management [34]. Moreover, bioactive compounds and nutraceuticals can be a first step for treating short clinical crows before resorting to the use of drugs, as they are able to exert innumerable potential biological actions.

Because HFD consumption is an inductor of MS, dietary interventions and targeted nutritional therapies could provide great promise for its prevention and treatment. Adopting a healthy lifestyle is already the cornerstone of MS management. Among the dietary interventions to reduce the incidence of MS, an increased intake of complex carbohydrates and lean proteins is recommended, as well as limiting the intake of saturated fat. In addition, increasing the consumption of fiber, even in the framework of a HFD, could alleviate many of its negative effects, especially those related to the intestinal barrier, where fiber improves the composition of the microbiome.

Regarding pharmacological strategy, poly-pharmacological therapy is often required [35]. The approach includes drugs that, in addition to their specific indications (hypoglycemic, hypolipidemic, hypotensive, etc.), combine their effect increasing insulin sensitivity in peripheral tissues. Hence, antihyperglycemic agents, such as pioglitazone or dipeptidyl peptidase-4 inhibitors, sitagliptin, have been shown to be beneficial in MS patients. More recently, liraglutide, a GLP-1 receptor agonist, has been investigated and used in clinical practice for its direct anti-atherosclerotic action. Likewise, statins have been studied for their anti-inflammatory effect by reducing blood levels of high-sensitivity C-reactive protein, and for their beneficial antithrombotic role in MS.

On the other hand, currently, the role of phytochemicals, bioactive compounds and nutraceuticals (plant extracts, spices, herbs and essential oils) is being widely investigated in the treatment of MS [36]. In need of further research, positive results have been obtained that make them a promising alternative for the development of new therapies. Among bioactive compounds, flavonoids stand out because of the large number of studies investigating their efficacy against MS. For example, the possible preventive role of soy isoflavones in metabolic syndrome-induced cardiovascular disease is worth mentioning. Another flavonoid of interest in ameliorating the signs of MS is quercetin due to its antihypertensive, antihyperlipidaemic, antihyperglycaemic, and other properties [37]. For more information, we recommend referring to the review by Gouveia et al. [38].

Within the group of flavonoids, proanthocyanidins (PAs), present in fruits and vegetables, have also been shown to have beneficial effects on the prevention of MS [29,39,40]. The consumption of PAs is commonly used for suppressing inflammation, to improve insulin sensitivity, and/or decrease fat deposition. Therefore, PAs seem to exert beneficial effects on metabolic disorders and prevent the onset of several oxidative stress-related diseases. In addition, dietary PAs and their metabolites, acting as prebiotics, lead to specific changes in GM composition and/or functions, which are beneficial to maintain the integrity of the intestinal barrier, and contribute to the host’s health [39,41,42]. Despite the extensive research on the consumption and effects of PAs, no conclusive results demonstrate if the intake can be useful to prevent or reverse MS by modulating GM and improving intestinal health. Some controversial results have been found, which may be related to the diversity of experimental designs, the timing of the inclusion of PAs, and/or duration of study.

Considering this, the results related to GM composition and intestinal parameters after PAs supplementation in animal models of diet-induced MS have been extensively analyzed in this review. Special attention has been paid to the timing of the introduction of PAs into the diet to better define the separate preventive and curative role of PAs in gut health in MS.

## 4. Proanthocyanidins

### 4.1. General Aspects

PAs, also called condensed tannins, are a subgroup of plant polyphenols that are widely distributed in the food we consume. They constitute the second most abundant group of phenolic compounds in diets of western countries, only behind lignins [43]. The greatest dietary sources of these compounds are legumes, cereals, nuts, cocoa, tea, wine, and fruits, such as blueberries, plums, grapes, and apples [13,44,45].

PAs are products of the secondary metabolism of plants synthesized by the general route of flavonoids, as part of the response to external aggressions. These compounds have been shown to have numerous benefits for human health [39,46]. Consumption of PAs is also associated with a lower risk of developing obesity and other disorders related to MS, although the full biological effects are not currently known [40]. Some of the multiple benefits attributed to PAs are antioxidant and anti-inflammatory activities, as well as immunomodulatory, anticancer, antimicrobial, neuroprotective, and cardioprotective effects [47]. PA intake has been widely studied and recommended in recent years as part of a varied diet or as supplements, due to the potential ability to prevent or reverse diseases associated with oxidative stress, as is the case of MS. Their antimicrobial and anti-inflammatory effects have been demonstrated in numerous viral and bacterial infections. More recently, PAs have also been observed to regulate the SCFA production [48]. However, despite being the subject of multiple studies, the specific role of PAs in preventing or reversing MS by maintaining or restoring intestinal homeostasis is not fully defined.

### 4.2. Structure and Classification

Based on their structure, PAs are included in the group of flavonoids, and within the subgroup of flavan-3-ols. The basic structure of flavonoids comprises two phenolic rings (A and B) separated by three carbons that form a third oxygenated heterocyclic ring. The degree of oxidation of this heterocycle determines the subcategory, as shown in the Figure 2.

PAs are formed by the polymerization of flavan-3-ol units, through a still partially unknown mechanism [46,49]. The two large groups of PAs arise depending on the bond that joins these subunits. In this way, in type-A PAs, flavanol monomers are linked by two bonds: a first carbon-carbon bond (interflavanic bond) between the C4 and C8 positions, or sometimes between C4 and C6, and a second carbon-oxygen bond between the C2 position and the hydroxyl of carbon 7 (C2 → O7) or, less frequently, between C2 and O5. However, type-B PAs only present the interflavanic bond between the C4 and C8 or C4 and C6 positions, being the first more frequent [5,13,43,46,49]. The most common forms of PAs are shown in Figure 3. PAs detected in food are often type B, whereas type-A PAs have only been found in curry, cinnamon, blueberries, peanuts, plums, and avocado [44].

Regarding their structure–activity relationships, some studies attribute their antioxidant capacity to their hydrogen-donating ability because they can end radical chain reactions and neutralize free radicals. It has also been related to different substitutions in the aromatic ring and side chain structure. Furthermore, the more hydroxyl radicals in the flavonoid nucleus, the better the antioxidant activity [50].

### 4.3. Absorption, Metabolism, and Excretion of Proanthocyanidins, Biotransformation by the Microbiota

The bioavailability of PAs is poor compared to other flavonoids and polyphenols due to their complexity and polymerized structure that makes them difficult to [13,51,52]. In vivo studies suggest PAs are not degraded in the stomach and reach the intestine with no relevant alteration [44]. PAs concentrations in intestinal lumen are a great deal higher than in the bloodstream [48]. Furthermore, in the small intestine, oligomeric PAs (dimers, trimers, and possibly tetramers) can be absorbed [13,52]. Because no transporters have been found in the small intestine for PAs, absorption occurs by passive diffusion through the paracellular pathway, because the numerous hydroxyl groups give it a hydrophilic character that is not suitable for a transcellular route [43,52]. In contrast, polymeric PAs (whose degree of polymerization is above four) are not absorbable due to their high molecular weight and polarity, which is why they reach the large intestine unchanged [53]. This characteristic makes its fermentation necessary to be absorbed and to exert systemic effects. In the cecum and colon, GM ferments unabsorbed PAs [54]. However, Choy et al. [55] and Liu et al. [56] found approximately 10% of the administered dose intact in feces. Bacterial fermentation results in low-molecular weight phenolic acids (which include phenylacetic, phenylpropionic, phenylvaleric, and phenylbutyric acids), and phenylvalerolactones. Both can cross the intestinal barrier [13,43,45]—Tao et al. [13] provides a more in-depth review of PAs fermenting GM and obtained metabolites. Therefore, PAs exert their systemic effects directly through the absorbed monomers/oligomers and bacterial metabolites of the polymeric structures. However, the local effects of PAs on the digestive tract should not be ignored. The intestinal effects derive from the presence of non-fermented PAs and from certain bioactive metabolites produced in fermentation and which are mainly active locally. Because these local effects on the digestive tract can also have systemic consequences, we highlight that PAs can also exert this intestinal mechanism that contributes to the systemic ones, without absorption being necessary.

### 4.4. Bidirectional Relationship of Proanthocyanidins with the Microbiota

A small percentage (approximately 10%) of ingested PAs are not absorbed and cross the colon to be eventually eliminated in the feces [55,56]. Although traditionally some argued that the effects of PAs were exerted by active metabolites capable of being absorbed and reaching different tissues, many recent investigations have shown the importance of the effect that PAs have on the intestine, so this effect may have repercussions at the systemic level. Therefore, absorption is not a necessary requirement for its biological activity as part of its effects derive from intimate contact with the intestinal barrier and modulation of the GM [41,52].

The composition of the GM is not steady; it varies with different factors such as age, health, diet, consumption of alcohol, tobacco, or antibiotics. Among the dietary components that can modify GM are PAs, possibly due to their antimicrobial [43] and prebiotic activities [13]. PAs show their actions by inhibiting the growth of some potentially pathogenic species to allow the development of beneficial species [41,57].

Likewise, GM is an indispensable factor in obtaining the active metabolites that fulfill the systemic PAs’ functions. Changes in the composition of the GM consequently affect the profile of the metabolites obtained, affecting PA activity [5,41,43]. Therefore, evidence shows there is a beneficial bidirectional relationship between the GM and PA consumption [5,49,52].

## 5. Gut Microbiota Modulation by Dietary Proanthocyanidins in Metabolic Syndrome

In the last decade, many researchers have opened the way to evaluate the role of PAs on health. Most studies evaluated PAs’ extracts from apple, blueberry, cocoa, carob, and grape seeds (a major source of PAs) [29,58]. The consumption of various sources of PAs has a wide variety of beneficial effects in maintaining or improving intestinal function. Therefore, it has become a promising strategy for the prevention or treatment of MS and other pathologies associated with intestinal dysfunction.

The intestinal benefits of PAs consumption derive from the modulation of the GM toward a healthier profile, which leads to changes in the production of SCFAs, and the maintenance or restoration of the integrity of the intestinal barrier. These effects promote colonic homeostasis.

### 5.1. Results Obtained from Rodent Experiments

Numerous studies have evaluated the impact of the consumption of PAs from foods or extracts on the GM of animals with diet-induced MS. A critical aspect of these studies is the timing of the introduction of PAs into the diet. In this sense, three alternatives are differentiated: (1) PAs are introduced before starting the consumption of the MS-inducing diet (t < 0) [59]; (2) PAs are introduced at the same time as the MS-inducing diet (t = 0) [12,56,60,61,62,63,64,65,66]; and (3) PAs are introduced into the diet when the pathology is already established (t > 0) [61,67,68]. However, in alternatives 1 and 2 (t < 0 and t = 0), PAs are used as a prevention strategy for MS; alternative 3 (t > 0) deals with a treatment strategy for MS. The difference between prevention and treatment is substantial. If PAs are administered as preventive (t < 0 and t = 0), their function will be to maintain healthy GM, avoiding the harmful or deleterious effects of the MS-inducing diet and the consequent dysbiosis that will ultimately contribute to the development and progression of MS. In contrast, if PAs are administered when the MS is already established (t > 0), it is foreseeable that its colonic effect will reverse, at least partially, dysbiosis. It should be noted at this point that PAs can have beneficial effects at the colonic or systemic level without acting on the GM. It is important to remember the bidirectional relationship between GM and PAs [13]. Therefore, in experiments that follow a treatment strategy (t > 0), the MS-inducing diet may cause changes in the populations that ferment PAs (some species of the *Lactobacillus*, *Clostridium*, *Eubacterium*, and *Bacteroides* genera [5]). When PAs are administered, they reach the colon and ferment, leading to metabolites that can differ from those produced in a physiological situation or in a prevention strategy (t < 0 and t = 0). These qualitative and quantitative changes in the production of active metabolites in the treatment strategy (t > 0) may have repercussions both on the local colonic effect and on the metabolic, antioxidant, and anti-inflammatory effects described at the systemic level. Thus, it is not surprising that the effect of prebiotic PAs could be different in treatment strategy (t > 0), when PAs are administered in animals with MS, than in prevention alternatives (t < 0 and t = 0), in which PAs are administered when there is no dysbiosis present. Therefore, studies must describe with maximum clarity whether what they perform is a prevention or treatment strategy, and that the discussion of their results is consistent with the particularities of the study design.

Table 1, Table 2 and Table 3 indicate changes in GM of experiments of different duration, conducted in several diet-induced MS animal models, after supplementation with PAs from different sources and doses, and introduced as preventive or treatment. As stated in the Introduction, PAs present a dual role: a prebiotic effect on beneficial species and an antimicrobial activity on potentially pathogenic species, driving GM toward a healthier profile. Thus, firstly, the prebiotic effect of PAs on bacteria considered beneficial in an MS situation is evaluated (Section 5.1.1); followed by its effects on bacteria which there is ignorance or controversy in the literature (Section 5.1.2); and, finally the antimicrobial effects of PAs on bacteria with harmful effects is described (Section 5.1.3).

#### 5.1.1. Effects of Proanthocyanidins Consumption on Bacteria Considered Beneficial in Metabolic Syndrome

Regarding changes related to bacteria that exert potential positive effects on health, PAs favored the growth of butyrate-producing bacteria (Table 1). Among them, *Roseburia* spp. stands out. *Roseburia* spp. are decreased due to Western diet consumption, but they are considered beneficial for intestinal health, having a positive impact by decreasing secretion of proinflammatory cytokines and improving intestinal integrity [12,69]. *Roseburia* spp. and the *Clostridium* XIVa cluster (which includes the *Roseburia* genus) maintain high levels after PA consumption in experiments by Liu et al. [56] and Van Hul et al. [12] in which, in both cases, mice were fed preventively HFD (t = 0) supplemented with extracts rich in PAs. The increase in other prominent butyrate producers such as *Allobaculum* sp. [12] and *Faecalibacterium prausnitzii* [61] has also been detected after PA intake as a preventive strategy (t = 0), compared with an MS control group. *Allobaculum* sp. is considered beneficial for intestinal health [69], and *F. prausnitzii*, which is one of the most abundant butyrate-producing bacteria in humans, exerts anti-inflammatory effects mainly locally in the colon [70]. Interestingly, Macho-González et al. [61] compared preventive vs. treatment PAs from carob-fruit extract consumption [alternatives 2 (t = 0) vs. 3 (t > 0)], and *F. prausnitzii* levels were only kept with PAs preventive intake (t = 0). In the treatment group (t > 0), PAs intake could not reverse the decrease in *F. prausnitzii* displayed in the MS control group. The results of Macho-González et al. [61] were evaluated in a T2DM model, although its inclusion in this review seems justified as rats also presented MS, as usual in T2DM animals, proven by dyslipidemia and a notorious non-alcoholic steatohepatitis.

Supplementation of PAs from MS-inducing diets caused other bacterial increases, non-butyrate producers, that are potentially beneficial for intestinal health (Table 1). A prominent case is the intestinal bloom of *Akkermansia muciniphila*, which shows rapid growth after PAs supplementation, positively related to intestinal health due to its anti-inflammatory and stimulating properties of mucin production [12,29,59,63,67,71,72]. *A. muciniphila* has been used as a probiotic, demonstrating its beneficial properties in cancer, diabetes, obesity, and inflammation [13,47]. Anhê et al. [59] (PAs as preventive, t < 0), Anhê et al. [67] (PAs as treatment, t > 0), and Rodríguez-Daza et al. [63] (PAs as preventive, t = 0) revealed an increase in *Akkermansia* spp. and *A. muciniphila* (Table 1), despite using different strategies (PAs introduction at t < 0, t = 0, or t > 0), different animal models of MS, and different PA origins. The comparison between preventive (t < 0) vs. treatment (t > 0) strategies can be found between the studies of Anhê et al. [59] and Anhê et al. [67]. Although the abundance of *Akkermansia* spp. was greater in the PA group than in the MS control in both preventive and treatment strategies, the changes were more pronounced when PAs were preventively used. Other bacteria in Table 1 include the phylum of *Bacteroidetes*, and *Coprobacillus* genus, because of their beneficial effects. According to Van Hul et al. [12] and Xiao et al. [64], a preventive consumption of PAs (t = 0) increased *Bacteroidetes* abundance. The phylum *Bacteroidetes* includes some necessary bacteria for fiber fermentation and is characteristically decreased in different metabolic disorders, such as obesity or T2DM [5]. However, *Coprobacillus* spp. increased in the study of PA treatment by Anhê et al. [67] (t > 0) and has been suggested as a genus capable of controlling the growth of pathogens such as *Clostridium difficile*, and favoring the growth of beneficial bacteria, for example *A. muciniphila*. Therefore, it is considered that *Coprobacillus* spp. can contribute to the generation of a healthy intestinal environment [73].

The *Bifidobacteriaceae* family (which includes *Bifidobacterium* genus) and the *Lactobacillales* order (which includes *Lactobacillus* genus) (Table 1) have been widely used as probiotics to improve gut health in various diseases, including MS [10,68,74]. These bacteria produce lactate and acetate as final fermentation products which, in a complex cross-feeding mechanism, are used by butyrate-producing bacteria (for in-depth information relative to cross-feeding, read the Louis and Flint revision [75]). According to Lee et al. [60], Macho-González et al. [61], and Zhu et al. [66], these taxa increased after PAs supplementation following a preventive strategy (t = 0). However, when PAs are introduced as treatment (t > 0) the results differ among studies. Xu et al. [68] detected that *Bifidobacteriaceae* and *Lactobacillus* spp. increase when *Pyracantha fortuneana* was included in the diet two weeks after the HFD. In contrast, Macho-González et al. [61] found no change in *Lactobacillus* and *Bifidobacterium* genera, not even an upward trend when the PAs of carob-fruit extract were incorporated as a treatment strategy (t > 0) three weeks after a high-fat high-cholesterol diet. Therefore, the discrepancy between the studies by Xu et al. [68] and Macho-González et al. [61] can be attributed to several factors, highlighting the composition of extracts and the MS progression when PAs are included in the diet. Regarding composition, the effects can differ depending on the PA richness of the extract, the polymerization degree of PAs, and the presence of other associated compounds that act synergically with PAs. However, while monitoring MS progression in the two commented studies, the results suggest that if MS is fully established [61], the prebiotic effect of PAs on *Lactobacillus* spp. and *Bifidobacterium* spp. could be compromised. Still, Macho-González et al. [61] demonstrated that rats already had clear fasting hyperglycemia when PAs were introduced as treatment, whereas in the study by Xu et al. [68], no confirmation of MS biomarker was reported.

#### 5.1.2. Effects of Proanthocyanidins Consumption on Bacteria Which There Is Ignorance or Controversy in Relation with Metabolic Syndrome

Table 2 collects the changes in bacteria modified by the consumption of PAs, but whose effect on intestinal health and the relationship with MS are not well established. Thus, the white color responds to a not well-established relationship, whereas gray color responds to a controversial relationship. First, bacteria with limited available evidence about their beneficial effects on MS are represented in Table 2 in white. Some authors have reported changes in bacteria of *Actinomycetales* and *Turicibacterales* order [68], *Lachnospiraceae* and *Ruminococcaceae* family [65], *Barnesiella* [67], *Bilophila*, *Lachnoclostridium* [64] *Oscillospira* [68], *Proteus* [64] and *Ruminococcus* [68] genera, or *Adlercreutzia equolifaciens* [62,63] and *Muribaculum intestinale* species [62] after PA intake. The administration of heat-killed *Actinomycetales* has been linked to the prevention of obesity and T2DM development in rats [76]. *Barnesiella* spp. seem to decrease after Western diet consumption, whereas their increase because of probiotics and prebiotics administration has been associated with better anti-inflammatory responses [77]. Moreover, Dudonné et al. [77] showed the *Barnesiella* genus could become a determining phyllotype to correct dysbiosis in inflammatory disorders. To date, no relationships have been established between *Turicibacterales*, *Lachnospiraceae*, *Ruminococcaceae, Bilophila* spp., *Lachnoclostridium* spp., *Oscillospira* spp., *Proteus* spp., *Ruminococcus* spp., *A. equolifaciens*, and *M. intestinale* with MS.

Contradictory information is available regarding the *Proteobacteria* phylum, the *Bacteroides* and *Blautia* genera, and *Blautia coccoides-Eubacterium rectale* and *Clostridium leptum* groups (Table 2, in gray). Liu et al. [56], Lee et al. [60], and later, Xiao et al. [64] referred to the increase in *Proteobacteria* phylum after PAs preventive consumption (t = 0). The authors defend that the marked increase in this phylum, specifically gamma *Proteobacteria*, was correlated with goblet cells proliferation and *Muc2* expression [60]. In contrast, in most publications, the *Proteobacteria* phylum has been associated with epithelial dysfunction, inflammation, and MS [78,79]. However, because this phylum is one with the greatest genetic variability [80], the determination of this phylum abundance is not conclusive, with the study of specific change in bacterial groups required.

The controversy found with the *Bacteroides* and *Blautia* genera (Table 2, in gray) is similar; therefore, the available results will be discussed together. *Bacteroides* is one of the main genera of *Bacteroidetes*. The *Bacteroides* genus includes saccharolytic bacteria that produce SCFAs from complex polysaccharides and fiber [71]. As indicated by Rios-Covian et al. [81], the consumption of a diet with sufficient protein and fiber would regulate the metabolism of *Bacteroides* spp. toward propionate production. As stated in the Introduction, propionate performs beneficial functions, which explains why, under physiological conditions, this genus is considered good for health [82]. In contrast, an increase in *Bacteroides* spp. in animals and patients with diabetes has been related to glucose metabolism alterations [83]. Consumption of unbalanced diets may be responsible for the loss of beneficial effects ascribed to the *Bacteroides* genus. The results in *Bacteroides* spp. in MS models are not uniform, as deduced from the Macho-González et al. [61], Liu et al. [56], and Xu et al. [68] studies. Although a rise in *Bacteroides* spp. was found in Macho-González et al.’s [61] preventive T2DM group (t = 0), a decrease in its abundance was reported in both preventive (t = 0) and treatment (t > 0) strategies performed by Liu et al. [56] and Xu et al. [68], respectively. Regarding the *Blautia* genus, increases in the abundance were detected in preventively supplemented groups (t = 0) by Liu et al. [56], Xiao et al. [64], and Zheng et al. [65], as well as in the PA-treated group of Xu et al. [68]. The *Blautia* genus plays a crucial role in glucose metabolism. In physiological situations, a higher abundance of *Blautia* spp. because of PAs consumption is considered positive [84]. However, in animals and patients with T2DM, the *Blautia* genus has been associated with the progression of T2DM and worse glycemic control [85,86,87], as well as weight loss with these bacterial decline [88]. Moreover, *Blautia* is the main genus included in the *B. coccoides-E. rectale* group. Regarding the consequences of PA consumption, in the study performed by Macho-González et al. [61], the *B. coccoides-E. rectale* group decreased in rats that consumed carob-fruit extract as preventive (t = 0) and treatment (t > 0) strategies against MS progression. In summary, PA consumption has been related to increases in physiological conditions in *Bacteroides* spp. and *Blautia* spp., which is considered beneficial for health [84]. However, some publications report an increase in both genera in MS and T2DM models induced by high saturated fat and high-protein diets [86]. This increase appears to promote the loss of integrity of the colonic barrier and worsen glycemic control [83]. Therefore, more studies are needed to evaluate the involvement of *Bacteroides* and *Blautia* genera in the development and progression of MS and T2DM, as a preliminary step to interpret the sometimes contradictory results on changes in their abundance after consumption of PAs [89].

The last group included in Table 2 is the controversial *C. leptum* group (gray). It includes a wide variety of bacteria—some included in Table 1, such as *F. prausnitzii*—which is a butyrate-producing bacteria that exhibits anti-inflammatory effects and helps maintain intestinal health [70]. The interpretation of the changes in this group after the consumption of PAs is complex. It would be necessary to determine parallel the abundance of *F. prausnitzii*, as it must be verified that the decrease in the *C. leptum* group is not due to a non-desired lower abundance of *F. prausnitzii*.

#### 5.1.3. Effects of Proanthocyanidins Consumption on Bacteria Considered Deleterious in Metabolic Syndrome

Table 3 indicates bacteria negatively related to gut health because they have been associated with infections, inflammation, and the development of MS. It is observed that the consumption of PAs reduces the presence of these bacteria, both in prevention and treatment strategies. Among the bacterial taxa whose presence was decreased by PAs supplementation are *Firmicutes* phylum, *Desulfovibrionaceae* and *Enterobacteriaceae* family, *Enterococcus* and *Lactococcus* genera, and *Desulfovibrio* genus (Table 3). Numerous bacteria with beneficial functions (e.g., *Clostridium* XIVa cluster, *F. prausnitzii*, *Lactobacillus* spp. or *Roseburia* spp.) are included in the *Firmicutes* phylum, decreasing in studies of Lee et al. [60], Liu et al. [56], Xiao et al. [64], and Zhu et al. [66], and should be studied independently, because changes at the phylum level are not representative of changes at lower levels [13]. Therefore, although *Firmicutes* must be considered to obtain the F/B ratio, it is of little interest as an isolated marker. The *Desulfovibrionaceae* family and the *Desulfovibrio* genus (declines reported by Van Hul et al. [12], Xu et al. [68], and Zheng et al. [65]) are characterized by their ability to reduce sulfate to hydrogen sulfide (H_2_S) and have been related to epithelial damage, impaired glucose tolerance, and MS development [12,68,90]. The *Enterobacteriaceae* family (less abundant in the Macho-González et al. [61] treatment group (t > 0)) includes numerous genera related to intestinal diseases with pathogens and opportunistic pathogens, such as *Escherichia*, *Klebsiella*, *Yersinia*, *Citrobacter*, *Shigella*, *Salmonella,* and *Serratia*. Furthermore, because these bacteria are Gram-negative (with LPS in the outer membrane), reducing their presence can have a positive impact on limiting endotoxemia [91]. The presence of *Enterococcus* spp. decreased in the preventive study (t = 0) of Zhu et al. [66] and in the treatment group (t > 0) of Macho-González et al. [61]. These bacteria are widely related to bacteremia because they are species with a high capacity to penetrate the epithelium and cause localized and systemic infections. Despite this, some strains have shown benefit when used in MS as probiotics [92]. Finally, *Lactococcus* spp. (declines, reported by Liu et al. [56] and Van Hul et al. [12], following preventive strategies, t = 0) are characteristically increased in metabolic disorders, such as MS and obesity, although their link with MS is not completely clarified [12,90,93].

Consumption of PAs decreased the F/B ratio in both preventive strategies (t = 0) of Zheng et al. [65] and Zhu et al. [66], and treatment strategies (t > 0) of Anhê et al. [67] and Xu et al. [68]. Taking these results into account, predictably, in the study by Macho-González et al. [61], the F/B ratio could also be reduced. However, the abundances of the *Firmicutes* and *Bacteroidetes* phyla, necessary to calculate the ratio, were not determined. Considering that the *B. coccoides-E. rectale* and *C. leptum* groups account for most *Firmicutes* present in feces [94], and that *Bacteroides* is the main genus of *Bacteroidetes*, an approximation of the F/B ratio can be made. Furthermore, the bacteria of these taxa (*B. coccoides* group, *C. leptum* group, and *Bacteroides* spp.) make up an average proportion of 96.3% (interquartile range of 89–99.2%) of the total of bacteria detected in feces of healthy and obese subjects in the study conducted by Schwiertz et al. [94]. Using this approach, the PA treatment group (t > 0) in Macho-González et al. [61] presented a lower ratio of *B. coccoides-E. rectale* + *C. leptum*/*Bacteroides* compared to the control MS group. Furthermore, the reduction of the F/B ratio was even more pronounced in the prevention group (t = 0).

The lower F/B ratio is associated with improved glucose tolerance in HFD-fed mice [95]. In contrast, a high F/B ratio is considered a characteristic marker of dysbiosis in MS. Often the *Firmicutes* phylum colonizes and infiltrates the mucus layer and has a much wider variety of metabolic pathways than *Bacteroidetes*, allowing them to extract more energy from a diet [71]. This results in increased production of SCFAs and therefore a greater contribution to host energy absorbed by the host, which promotes the development of obesity and other metabolic disorders [96]. In contrast, *Bacteroidetes* are located primarily in the intestinal lumen because their ability to interact directly with the intestinal wall is more limited. Furthermore, they can extract less energy from a diet due to the lesser variety of metabolic pathways [71].

#### 5.1.4. Relevant Considerations and Recommendations

PAs supplementation of an unhealthy diet could shift GM toward a more beneficial profile. Among the studies in Table 1, Table 2 and Table 3, only three were approached with a therapeutic strategy (Anhê et al. [67], Xu et al. [68], and Macho-González et al. [61]). Together, their main results were an increase in *A. muciniphila*, *Bifidobacteriaceae*, *Coprobacillus* spp., and *Lactobacillus* spp., as well as a reduction in *Enterobacteriaceae*, *Enterococcus* spp., *Desulfovibrionaceae*, and the F/B ratio. Therefore, PAs can be potentially useful natural compounds to prevent and treat MS-associated dysbiosis, but evidence suggests their consumption has less pronounced effects on the therapeutic strategy, when the disease has already settled, than on the preventive. Further studies comparing both strategies would be necessary to confirm this hypothesis.

Besides the changes in Table 1, Table 2 and Table 3, the experiments performed by Masumoto et al. [14] and by Liu et al. [97] must be especially mentioned. Masumoto et al. [14] fed mice HFD supplemented with apple-PAs (0.5%) following a prevention strategy (t = 0) for 20 weeks, the longest study period compared to the other studies in Table 1, Table 2 and Table 3. In the GM analysis, they found increased strains belonging to *Bacteroidetes*, *Bacteroides*, *Roseburia*, *Akkermansia*, and *Adlercreitzia*; and a decrease in those ascribed to *Firmicutes*, all consistent with the results represented in Table 1, Table 2 and Table 3. An increase in *Anaerovorax* sp. was also detected. This butyrate-producing bacterium [98] is notably reduced in patients with obesity [99], but there is no available information on its behavior in relation to MS [95]. In contrast to other studies, Masumoto et al. [14] detected decreases in *Bifidobacterium* spp., *Clostridium* spp., and *Laschnospiraceae*. These results are discordant with later evidence, as already highlighted by Liu et al. [56]. However, contradictory results have also been found in Liu et al. [97], who used a T2DM mouse model similar to Macho-González et al. [61] induced by HFD for four weeks and streptozotocin. These researchers evaluated GM composition by next-generation sequencing after a 4-week treatment strategy (t > 0), supplementing HFD with three different doses of a peanut skin extract (mainly containing A-type PAs). The results demonstrate increases in the F/B ratio, the *Bacteroidaceae*, *Lachnospiraceae*, and *Ruminococcaceae* families, the *Lachnospiraceae*_NK4A136 group (potential “new generation” probiotics [100]), and *Alloprevotella* genus after the consumption of peanut skin PAs.

Unfortunately, we have realized that contradictions and lack of information are common when comparing GM studies. The discrepancy may be due to the lack of homogeneity in the experimental conditions and methods used by the different authors. These are the main parameters that vary between published experiments: (a) animal models, sex differences, and MS-inducing diets; (b) PAs [origin, type (A or B) and dose], experimental scheme (preventive t < 0 and t = 0, or treatment t > 0), and experimental length; (c) sample collection [type of sample (dry feces, distal colon feces, caecum content) and state (fasting or postprandial)]; (d) analytical methods (next-generation sequencing methods or q-PCR); and (e) data analyses (bioinformatic or not). Consequently, this review proposes a homogeneous criterion to facilitate the comparison between studies.

Added to the animal model and the diet composition, authors should specify the specific experimental scheme, highlighting when PAs are introduced. This determines the nutritional strategy used, that is, if dietary supplementation is approached as a preventive or treatment of MS. Thus, if PA and MS-inducing diets begin simultaneously, it should be viewed as a prevention model, because the disease is not established when PAs are introduced. However, it should be a treatment strategy if PAs are introduced after the unhealthy diet has been consumed for a reasonable period for MS establishment. It is recommended that MS was confirmed before introducing PA, measuring a recognized variable such as fasting glucose [61,97], to ascertain treatment strategy.Most of the preclinical studies use male rodents. It is urgent that females are also included in preclinical studies, because MS-inducing diets have a different impact depending on sex. Furthermore, there are variations in the composition of GM between sex [101] that will determine a modified fermentation of PAs, crucial for their systemic effects. Last, due to the recruiting of men and women, it would be important to predict differences in PAs effects from animal experiments.As the type of PAs and the presence of associated compounds could relate to understanding their effects and the differences found in previous studies, it would be helpful for the authors to carefully describe the composition of PA extract used in each experiment.It is also recommended to use feces collected directly from the colon at the animal’s slaughter time. Thus, contamination can be avoided as much as possible, and the GM present in the sample will be representative.The presence of bacterial taxa strongly associated with the consumption of PAs or MS should be determined. It is increasingly common to perform next-generation sequencing of GM, leading to report a wide variety of changes in bacterial taxa whose role in physiological conditions or their relationship with MS is unknown [65,97,98,99,100,101,102,103,104]. Although this is positive, because it increases the knowledge about the changes in GM promoted by PA consumption in pathological conditions, the dispersion and variety of data makes their analysis difficult; and their poor knowledge complicates the interpretation. Therefore, to have the greatest evidence available, we suggest determining as a priority the abundance of strains ascribed to *A. muciniphila*, *Bifidobacterium*, *Desulfovibrionaceae*, *F. prausnitzii*, *Lactobacillus*, *Lactococcus*, and *Roseburia*. Furthermore, the abundance of *Firmicutes* and *Bacteroidetes* phyla can be measured to obtain the F/B ratio, although their analysis at the phylum level is not conclusive. It is recommended that results about these bacteria can be found in the manuscript, even if they have not changed after PA consumption.

### 5.2. Results Obtained in Clinical Trials

Although preclinical studies analyzing PA effects are abundant, it is usually reported in reviews there are not enough clinical trials to draw conclusions about PA efficacy in modulating GM in patients with MS. To verify this, we conducted an exhaustive search on ClinicalTrials.gov using parameters such as “gut microbiota,” “proanthocyanidins,” and “metabolic syndrome”; later we expanded the results with the terms “flavonoids,” “polyphenols,” “grape,” and “type 2 diabetes mellitus.” These searches resulted in 13 clinical trials in a “completed” status, and the other six in an active state. It is not surprising that there are numerous clinical trials underway, because nutritional intervention with polyphenols as prevention or treatment of metabolic diseases is an active line of research. Since 2020, the number of completed clinical trials has almost doubled (from 7 to 13 trials). However, focusing on the 13 completed clinical trials, the limited publication of the results is striking. Six (NCT01944579, NCT02407522, NCT02728570, NCT03800277, NCT03754504, and NCT04130321, all with treatment strategies) have not yet published any results. Reasons may be diverse, highlighting the lack of time to process the results or that the results did not satisfy the hypothesis of the study. Among the seven studies that have published part of their results, five (NCT03523403, with a prevention strategy; NCT01010841, NCT04075032, NCT04075032, and NCT04130321, with treatment strategies) have not published GM data despite reporting they were analyzed. Therefore, only two clinical trials have published the results on changes in the GM after the consumption of foods with PAs. The first clinical trial was NCT03076463 and comprised supplementing grape pomace in patients who met at least two of the three criteria for the diagnosis of MS. Most patients met all the requirements for the diagnosis of MS at the time of inclusion in the trial. Therefore, it has been considered that they used a treatment strategy. Grape pomace supplementation reduced plasma insulin concentrations and the HOMA index in the responder group, although it produced no change in their GM composition. In contrast, in the non-responder group (patients whose insulin resistance did not improve), they detected an increase in *Bacteroides* spp., a genus that appears to be related to poor glycemic control [105]. The other clinical trial with published results is Direct-plus (NCT03020186), conducted in patients with obesity or dyslipidemia over 30 years of age. Therefore, it is a prevention strategy. Furthermore, it should be considered that the Direct-plus trial analyzed the effect of a modification of the usual diet with an increase in polyphenols and a reduction in red meat. Therefore, it is difficult to establish the possible isolated effect of PAs. The results obtained show that the consumption of a Mediterranean diet rich in polyphenols modifies the GM of the patients. Numerous changes were detected, mainly in non-core taxa (lower than 50% prevalence), highlighting the marked increase in *Prevotella* spp. and a decrease in *Bifidobacterium* spp. abundance [106]. Considering these two clinical trials and the exhaustive analysis in animal models of MS, these data suggest PAs are reduced in their ability to modulate GM when MS and the associated dysbiosis are established. In this review, we show that studies of animals have been systematically published with prevention strategies that erroneously generalize the efficacy of PAs in the treatment of MS. This could have encouraged clinical trials in patients with clearly established MS, without there being apparently sufficient preclinical evidence to do so. Many of these clinical trials have not published their results, which may be because of the lack of significant changes (*p* > 0.05). We underline the urgency for non-significant results from clinical trials to also be published to contribute to general knowledge and avoid duplication of further studies. The greatest benefit of PAs consumption is likely to be achieved in patients who, although they are at risk of developing MS, do not meet the three altered parameters required to be diagnosed. The establishment of MS own dysbiosis may be one cause of PAs partially losing their prebiotic effect, widely demonstrated in a physiological situation.

## 6. Restoration of Intestinal Epithelial Barrier Integrity by Dietary Proanthocyanidins

This section focuses on literature related to the effects of supplementing PAs on the loss of integrity of the intestinal barrier promoted through HFD consumption and associated with MS. We focus on the mechanisms through which HFDs—the Western diet or the cafeteria diet—and their consequent dysbiosis cause the loss of the integrity of the intestinal barrier, allowing the entry of toxic products into the blood.

An increased intestinal barrier permeability produces exaggerated immune responses and favors the development of diseases that cause intestinal inflammation, as occurs in MS. The activation of the intestinal immune system has recently been reviewed in-depth by Andersen-Civil, Arora, and Williams [107]. Because inflammation and oxidative stress are described to occur simultaneously and feed on each other, their separation is meaningless. Therefore, compounds with anti-inflammatory and antioxidant properties, such as PAs, can slow down the intestinal barrier dysfunction and should be useful in the prevention or treatment of MS. Therefore, this section reviews the effects of PAs on (a) the production of mucus and antimicrobial peptides; (b) the morphology of the intestinal wall; (c) the permeability of the intestinal barrier; and (d) the activation of the intestinal immune system and the production of reactive oxygen species (ROS). Because typical alterations in these points lead to the loss of intestinal homeostasis in MS, their consideration is critical to properly evaluating the effect of PAs at the intestinal level.

### 6.1. Impact on Mucus and Antimicrobial Peptides’ Production by Dietary Proanthocyanidins

The first component of the intestinal barrier, lining the epithelium, is the mucus layer. Mucus prevents microbial invasion of the epithelium and subsequent inflammation, both events related to the development of MS. In addition, the antibacterial peptides secreted by Paneth cells are retained in this layer, which are attributed to a relevant role in the maintenance of homeostasis [26]. It is essential to maintain adequate thickness of the mucus layer and optimal antibacterial peptides’ production, which can reduce the exposure of the mucosa to the microbiota. However, it can be affected by HFDs and dysbiosis associated with MS.

Various studies have been conducted in diet-induced MS models to evaluate the preventive effect of PAs consumption protecting the mucus layer. First, type-A PAs from different species of blueberry increased the number of goblet cells (mucus-producing) per crypt [60,63], the mRNA expression of *Muc2* in ileum [60], and the thickness of the mucus layer [63]. Furthermore, type-B PAs from carob-fruit supplementation increased the number of goblet cells per crypt in the distal colon [61]. These promising results are important in MS prevention because it has been demonstrated in rodent models of MS that the differentiation rate and *Muc2* transcription of goblet cells are reduced, resulting in a thinner mucus layer [60,63,108].

In contrast, there were no significant differences in the treatment group (t > 0) regarding the MS control, highlighting once more the importance of remembering the experimental design to understand the effects of PAs [61]. Furthermore, the treatment study carried out by Anhê et al. [67] showed that fecal mucin was greater in the PA-supplemented group vs. the MS control, although the increase was not significant. Therefore, PAs supplemented in a preventive strategy (t ≤ 0) could exert a direct protective effect on the mucus layer, which added to some changes previously described in the Section 5.1 (a decrease in *Firmicutes* and an increase in *A. muciniphila*) can favor the maintenance of the integrity of the barrier, preventing or delaying the development of MS. In treatment strategies, there is not enough evidence to ensure such PA effect.

In relation to PA effects in antimicrobial peptides’ production, also altered in MS, two preventive experiments in HFD-induced MS have been conducted. Preventive supplementation with PAs (t = 0) increased transcription of genes that codify antibacterial peptides. The intake of blueberry powder increased *Defb2* in mice [60], a cinnamon extract increased lysozyme-1, and a grape extract maintained the expression of *Reg3γ* at normal levels [12]. These increases in antibacterial peptides contributed to maintaining the integrity of the mucosal barrier [12,60], therefore, favoring intestinal homeostasis. Unfortunately, there are no studies that evaluate the treatment strategy (t > 0). It must be considered that only PA-ascribed effects can be demonstrated in those experiments that use pure extract.

### 6.2. Restoration of Intestinal Wall Morphology by Dietary Proanthocyanidins

Some researchers have evaluated the changes in intestinal morphology within the context of HFD consumption in MS models and the consequences of the inclusion of PAs in such diets. In these rodent experiments, HFDs reduced the size of the villi in the small intestine [60] and produced a quick inhibition of the proliferation of colonocytes in the colon [109]. This explains Zou’s et al. [110] results, who found a significant reduction in the mass and length of the colon, and atrophy of the crypts in the mice with MS. However, dietary supplementation with blueberry powder (rich in PAs) maintained a normal length of villi in the small intestine of mice fed a HFD comparable to those on a normal diet ([60]. Gao et al. [97] showed that preventive grape seed proanthocyanidins extract (GSPE) consumption (t = 0) reduced the crypt atrophy caused by HFD. Furthermore, supplementation with a carob-fruit extract maintains a higher crypt depth in the distal colon of the preventive group (t = 0) vs. the MS control group [61]. In contrast, controversial changes have been found when PAs were introduced as a treatment strategy (t > 0). Although Liu et al. [97] reported higher intestinal villus length and colonic crypt depth, Macho-González et al. [61] found no changes after dietary supplementation with PAs. These results support the need for the characterization of PAs extract, because discordant results may be due to different origins, types, or doses of PAs, despite similar MS-experimental models used.

### 6.3. Improvement of Altered Intestinal Permeability by Dietary Proanthocyanidins

Intestinal permeability is the ability of the intestinal mucosa surface to be selectively penetrated by substances. Microorganisms and other luminal antigens can passively cross the intestinal barrier through the space between epithelial cells. This flow is physiologically regulated by TJs, which are multi-protein complexes that bind adjacent epithelial cells and are essential to maintain this function [26]. However, MS dysbiosis, HFD, other unhealthy dietary components, oxidative stress, and proinflammatory cytokines, among other factors, can cause an abnormal/excessive increase in intestinal barrier permeability.

Some studies evaluate the effect of PA consumption on the maintenance (prevention design, t ≤ 0) or restoration (treatment design, t > 0) of intestinal barrier integrity. To analyze this, different approaches can be used, as follows. a) In vivo experiment: oral ovalbumin administration (OVA) test; b) ex vivo experiments: transepithelial electrical resistance (TEER); and c) in vitro experiments: c1-TJs analyses (gene expression, proteins levels, or immunolocalization) and c2-plasma LPS concentrations. Although some articles report the effects of PAs on TJs, this review does not focus on this aspect. The main reason is that the techniques used are diverse (PCR, Western Blot, ELISA, and immunohistochemistry), and discussing results thoroughly could be a matter of another publication. As a conclusion about TJ results, both in prevention [12,61,102,111] and treatment strategies [14,61,68,97,112,113], PA consumption strengthens TJs, mainly increasing the expression or levels of claudin and occludin. González-Quilen et al. [29] give more data about PAs’ modulating effects of TJs.

Regarding the effect of PAs on intestinal permeability, Gil-Cardoso et al. [111] designed a very different preventive experiment in an MS model induced by a high-fat high-non-complex carbohydrates diet (HFHNCCD). The GSPE preventive effect was evaluated in two groups (one with a t < 0 strategy and another with a t = 0 strategy). Both groups showed lower small intestine and colon permeability (especially in the duodenum) than the MS control after 12 and 17 weeks according to OVA test, TEER, and plasma LPS results. Feldman et al. [114] and Gao et al. [97] found similar results after a preventive strategy (t = 0). They reported lower serum levels of LPS after the consumption of PAs-rich extracts by animals with HFD-induced MS. However, between treatment strategies another experiment by González-Quilen et al. [113], analyzed the capacity of high concentrations of GSPE (between four and twenty times higher than the dose used in the preventive experiment) to reverse the alterations promoted by consumption of a HFHNCCD for 17 weeks. Grape seed PAs were introduced in the 15th week as for the treatment strategy (100 or 500 mg of GSPE per kg body weight). Rats treated with 100 mg/kg of GSPE showed lower intestinal permeability than the MS control group according to the OVA test, TEER, and plasma LPS concentrations. Likewise, Liu et al. [97] found lower serum levels of LPS in T2DM rats fed with HFD. Therefore, the consumption of PAs can reduce intestinal permeability, partially due to TJs strengthening [29], thus avoiding bacterial translocation and endotoxemia. Thus, PAs are potentially useful compounds to maintain or restore the integrity of the intestinal epithelium in the context of MS. The main intestinal effects of PA consumption on the regulation of intestinal permeability are summarized in Figure 4.

### 6.4. Improvement of Intestinal Inflammation, Immune Response, and Oxidative Stress by Dietary Proanthocyanidins

Dietary management against intestinal inflammation and oxidative stress associated with MS could be a decisive aspect in its prevention and treatment. Both factors promote early damage to the intestinal barrier, participating in the initial stages of MS; and immediately, when the intestinal barrier is altered, new inflammatory pathways are activated, and there is a higher production of free radicals establishing a dangerous vicious circle. Among the pathways predominantly evaluated, which are more frequent tested on the colonic barrier than on the small intestine, luminal antigens stand out, with LPS as the predominant bacterial antigen, because of activating the immune system through TLRs. LPS binds to TLR4 and activates MAPK and transcription factor NF-κB pathways, leading to loss of intestinal barrier integrity, immune cell infiltration, and oxidative stress [29,90].

It has been extensively demonstrated that PAs reduce different inflammatory mediators in several tissues and MS; numerous animal experiments have been conducted in MS models to evaluate their anti-inflammatory effects on the intestine. Four preventive assays (t ≤ 0) with diet-induced MS in rodents showed a reduction in the proinflammatory cytokines and inducible enzymes expression. In the study by Anhê et al. [59], mice that received red cranberry PAs had lower ciclooxigenase-2 (COX-2) and TNF-α protein levels in jejunum than those fed a HFHNCCD. Furthermore, Lee et al. [60] reported that TNF-α mRNA expression in ileum was markedly lower in the blueberry-administrated group compared to its counterparts fed a HFD, keeping levels equal to the healthy group fed a low-fat diet. Likewise, GSPE administration as a preventive strategy reduced plasmatic TNF-α levels induced by HFHNCCD [111]. Moreover, preventive supplementation with GSPE (t = 0) blocked the increase in IL-1β, IL-6, and NF-κB mRNA levels in the ileum of HFD-fed rats [97]. However, treatment assays (t > 0) in rats with MS induced by a HFHNCCD supplemented with GSPE also showed less intestinal inflammation, less gene expression of proinflammatory cytokines such as IL-1β and inducible enzymes such as inducible nitric oxide synthase (iNOS) in the ileum [112], and less TNF-α secretion by the duodenum and colon [113]. Therefore, it appears that PAs reduce the production of mediators and enzymes, which participate in the immune response, in both preventive and treatment designs, thus blocking the inflammatory pathways and reducing the over-activation of the inflammatory cascade.

We found some specific mechanism that may be responsible for anti-inflammatory effects of the consumption of PAs. The first proposed is related to the potential ability of PAs to interact directly with LPS. Delehanty et al. [115] reported that cranberry PAs bind to LPS, thus preventing TLR4 activation. Furthermore, Xing et al. [116] suggested that procyanidin B1 acts as a competitive antagonist for TLR4, decreasing LPS binding. In both cases, the PAs blocked the pathway from the beginning. We highlight that polymeric PAs can conduct this action without the need to be fermented. Therefore, this could partially explain the satisfactory results obtained in treatment experiments, similar to preventive ones. However, this mechanism is not completely accepted and there is controversy, as some researchers reject the idea that PAs can interfere with LPS-TLR4 interaction, at least in dendritic cells [117,118].

Another mechanism involved with the anti-inflammatory properties of PAs affects downstream of the MAPK pathway, activated (among others) after LPS-TLR4 binding, which regulates the transcription of proinflammatory cytokines and MLCK enzyme [29]. Moreover, NF-κB activation, through the MAPK pathway or IκB kinase (IKK) complex, is a key factor causing the loss of intestinal integrity because of TJs disruption. PAs have been abundantly reported to hinder the activation of NF-κB, its translocation to the nucleus, and block its binding to DNA [29,119], consequently reducing the transcription of proinflammatory cytokines, chemokines, adhesion molecules, and inducible enzymes such as COX-2 and iNOS [120]. Thus, a lower activation of these pathways also entails the lower production of free radicals, preventing oxidative stress.

Regarding antioxidant properties, many studies have shown that PAs can reduce oxidative stress. PAs exert their antioxidant activity through different mechanisms:Radical scavenging activity, due to its aromatic structure [107,121]. PAs can prevent free radical-induced oxidative damage and avoid lipid peroxidation, which increases in MS. Again, polymeric PAs can exert this action independently of the degree of fermentation, ensuring positive results in treatment strategies (t > 0).Inhibition of the activity of enzymes that cause oxidative stress, such as neutrophil myeloperoxidase (MPO) over-activated in MS [111,112,113]. This enzyme produces large amounts of ROS to cause oxidative damage to invading microorganisms. However, overproduction of ROS over time depletes antioxidant defenses, causes oxidative stress, and has a negative effect on the organism. Ileal MPO activity was decreased by GSPE administration in both preventive [111] and treatment strategies for diet-induced MS, placing these values in the normal range [112,113].Inactivation of signaling pathways related to oxidative stress and an increase in the expression of antioxidant enzymes. As indicated above, PAs decrease the transcription of inflammatory enzymes that produce oxidative stress such as COX-2 and iNOS [113,122]. In addition, PAs regulate these signaling pathways toward activation of the transcription factor nuclear factor erythroid-2-related factor 2 (NRF2) [29,107,119,122,123], which favors the transcription of antioxidant enzymes (e.g., catalase, superoxide dismutase, glutathione peroxidase, etc.), thus strengthening the ability of tissue to defend against oxidative stress.

Figure 5 summarizes the mechanisms through which PAs exerts its anti-inflammatory and antioxidant activities.

The intestinal inflammation and oxidative stress characteristics of MS have also been related to the initiation and promotion of long-term carcinogenesis. Due to the low-grade chronic inflammation, aberrant crypt foci—which are tumor cells initiating carcinogenesis—can arise and lead to polyps [124]. Epidemiological and experimental studies confirm that patients with MS have a higher risk of developing colorectal cancer [125]. Therefore, the ability of PAs to regulate the cell cycle and apoptosis in the colon has been widely studied. In human cancer cell assays (CaCo2, HT-29, and HTC-116), PAs could arrest the cell cycle and induce apoptosis [126,127,128]. In contrast, in colonocytes, PAs did not induce cell death [126]. Far from inducing apoptosis, PAs positively regulated the antioxidant system in these non-tumor cells by reducing oxidative stress and consequently decreased apoptosis [121,129]. Macho-González et al. [61] used an MS/T2DM rat model, both preventive and treatment groups, fed with carob-fruit extract, and increased both proliferation and apoptosis in distal colonocytes, which could indicate a greater cell turnover. Therefore, PAs appear to exert a protective function against oxidative stress in colonocytes, and PAs have antiproliferative and proapoptotic activity on tumor cells that prevents the development of different types of colorectal cancer. Furthermore, the dual effect of polyphenols depending on cell type was reviewed by Wan, Co, and El-Nezami [130]. Thus, polyphenols have actions that promote cell survival in healthy cells and promote tumor cells death. This duality may be due to differences in polyphenols oxidative behavior. Therefore, polyphenols are considered useful compounds in the so-called “oxidative therapy.” This therapy is characterized by antioxidant effects on normal cells and pro-oxidant effects on cancer cells. Because PAs have antioxidant properties before being fermented by the intestinal microbiota, it is possible that the effects derived from such capacity (e.g., cell turnover) are maintained during the treatment strategies, as has been found in Macho-González et al. [61].

The epigenetic regulation of gene expression is another important mechanism involved in modulating the immune response, inflammation, and oxidative stress. Environmental factors, such as diet and physical activity, can produce epigenetic changes reflected in metabolic health [57,121]. A review has been published that evaluates the therapeutic potential of flavonoids for their ability to regulate cancer epigenetically [131]. However, there is a great ignorance of epigenetic actions of the PAs. To date, no studies have evaluated the effect of PAs consumption on epigenetic regulation in intestinal cells. However, in rats the epigenetic regulatory effect of polyphenolic fractions (including PAs) in the colonic mucosa has been tested, finding an epigenetic regulation of the immune and inflammatory function of the intestinal mucosa. Thus, the use of plum polyphenols reduced oxidative stress and inflammation of the colonic mucosa, while increasing the expression of microRNA-143. This microRNA-143, besides suppressing the inflammatory pathway mediated by protein kinase B (AKT), is considered antioncogenic due to its tumor suppressing activity [132]. Epigenetic regulation is a key factor in modifying pre- or post-transcriptional cellular responses that include changes in gene expression. Patients with MS have altered epigenetic regulation of genes related to proinflammatory mediators and lipid metabolism [121,133]. Therefore, it would be important to evaluate the ability of PAs to modify the epigenetic component of metabolic disorders. Its understanding would contribute to designing more efficient strategies supplementing PAs to prevent or treat MS.

## 7. Conclusions

Scientific evidence from reviewed studies demonstrates that maintaining or restoring of intestinal homeostasis can be decisive in preventing or treating MS, and supplementation with PAs stands out as an interesting nutritional strategy for managing MS. This review has focused on the intestinal effects of PAs on the frame of MS, emphasizing the alterations promoted by a HFD consumption. Among PAs’ effects, the microbiota composition and those on intestinal barrier integrity are highlighted. PAs can modulate the microbiome toward a healthy profile, increasing bacteria with beneficial functions for the host, while reducing the abundance of detrimental bacteria. To date, published clinical trials to verify preclinical findings are scarce. Therefore, PAs directly affect intestinal barrier integrity mainly through mechanisms that involve mucus and antimicrobial peptides production, strengthening barrier integrity, and anti-inflammatory and antioxidant properties. The comparison between papers that evaluate PAs’ effects is hindered by different aspects carefully analyzed in this review (different animal models, methodologies, wide ranges of time, dose, and origin of PAs, etc.). Experimental design is mentioned in particular. That is, when PAs are incorporated into the diet, determining if it is a preventive or treatment strategy is important. Because PA fermentation in the colon depends on GM composition, the establishment of characteristic dysbiosis in MS is a key event that conditions PAs’ intestinal and systemic beneficial effects. This review allows a systematic validation of the diverse effects of the PAs on the intestinal dysfunction in MS, differentiating between preventive and therapeutic actions. It can also be suggested that the preventive consumption of PAs in MS-associated dysbiosis and intestinal dysfunction induced by HFD is more successful than the treatment strategy. The therapeutic effect of PAs seems less pronounced in most of the reviewed aspects. Further studies comparing both strategies are necessary to confirm these hypotheses, not only in MS but also in its associated pathologies.

## Figures and Tables

**Figure 1 ijms-24-05369-f001:**
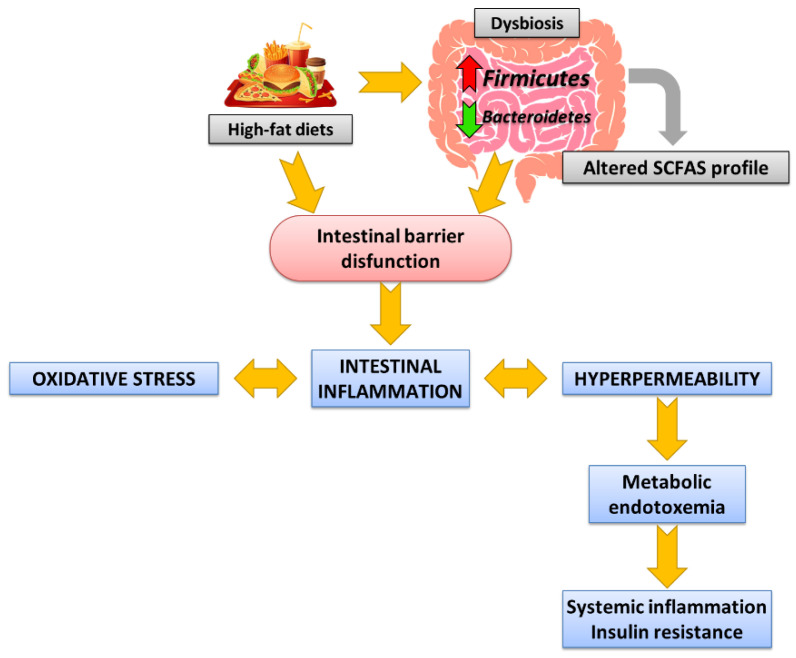
Internal connections between Western dietary intake, dysbiosis, and intestinal barrier dysfunction in MS development.

**Figure 2 ijms-24-05369-f002:**
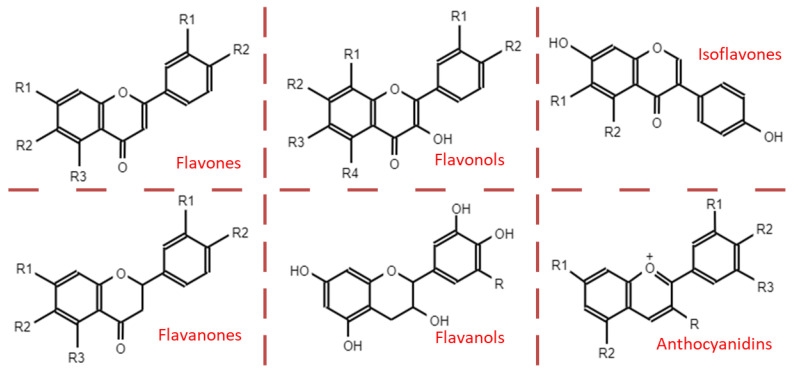
Main subcategories of flavonoids.

**Figure 3 ijms-24-05369-f003:**
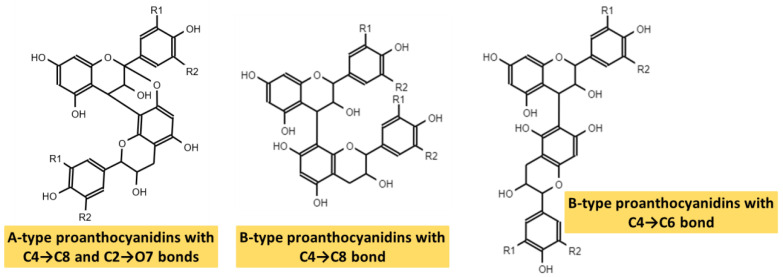
Types A and B of proanthocyanidins.

**Figure 4 ijms-24-05369-f004:**
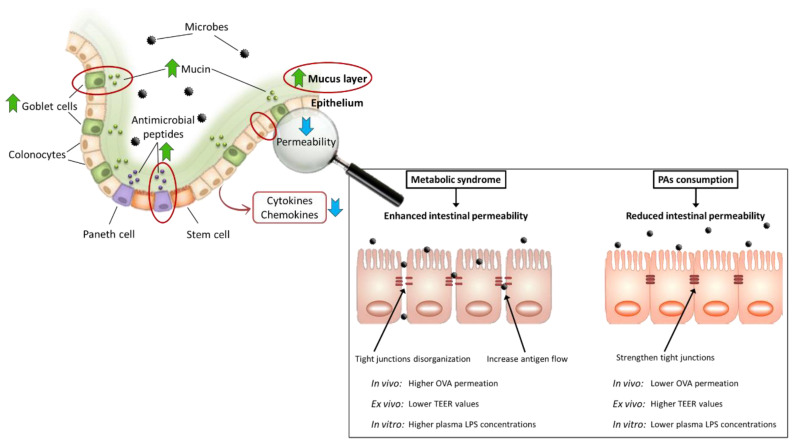
Intestinal regulation of intestinal permeability by dietary proanthocyanidins. ↑ increase; ↓ decrease.

**Figure 5 ijms-24-05369-f005:**
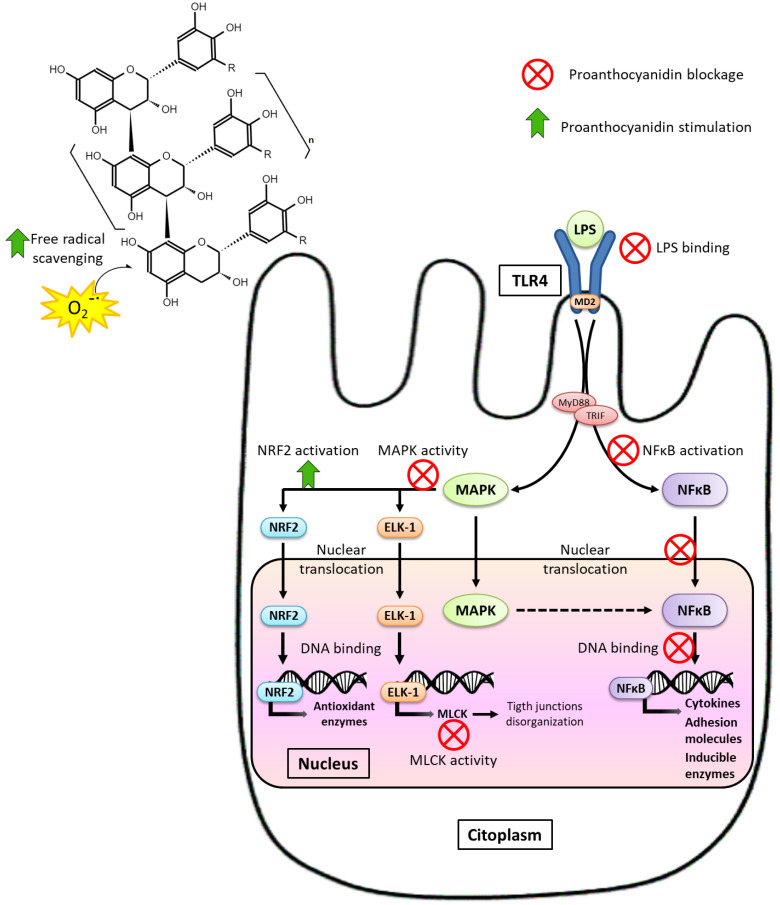
Mechanism involved in the anti-inflammatory effect of PAs.

**Table 1 ijms-24-05369-t001:** Effects of proanthocyanidin consumption on bacteria considered beneficial in metabolic syndrome.

Bacteria	Taxa	Phylum	Effect	Origin and Dose	Experimental Model	PA Introduction	PA Intake Period	Reference
*Akkermansia*	Genus	*Verrucomicrobia*	↑	Red cranberry PAs (300 mg/kg)	HFHNCCD mice (65% lipids, 15% proteins and 20% CH)	1 week before starting the HFHNCCD	8 weeks	[59]
*Akkermansia muciniphila*	Species	*Verrucomicrobia*	↑	Wild blueberry PAs (200 mg/kg)	HFHNCCD mice (65% lipids, 15% proteins and 20% CH)	Simultaneous beginning with the experimental diet	8 weeks	[63]
*Akkermansia muciniphila*	Species	*Verrucomicrobia*	↑	Cranberry extract PAs (200 mg/kg)	HFHNCCD mice C57BI/6J (65% lipids, 15% proteins and 20% CH)	13 weeks after HFHNCCD	8 weeks	[67]
*Allobaculum*	Genus	*Firmicutes*	↑	Cinnamon (2 g/kg) and grape extract (8.2 g/kg)	HFD mice (60% kcal fat diet)	Simultaneous beginning with the experimental diet	8 weeks	[12]
*Bacteroidetes*	Phylum	*Bacteroidetes*	↑	Cinnamon (2 g/kg) and grape extract (8.2 g/kg)	HFD mice (60% kcal fat diet)	Simultaneous beginning with the experimental diet	8 weeks	[12]
*Bacteroidetes*	Phylum	*Bacteroidetes*	↑	B2 procyanidin (0.2% of diet)	C57BL/6 mice with HFD (34% fat diet)	Simultaneous beginning with the experimental diet	8 weeks	[64]
*Bifidobacterium*	Genus	*Actinobacteria*	↑	Carob fruit extract (4 g/kg restructured meat)	HFD rats (50% fat, 1.4% cholesterol and 0.2% cholic acid)	Simultaneous beginning with the experimental diet	8 weeks	[61]
*Bifidobacteriaceae*	Family	*Actinobacteria*	↑	*Pyracantha fortuneana* (0.4% of diet)	HFD rats (Normal diet + 12% fat)	2 weeks after HFD	8 weeks	[68]
*Clostridium XIVa*	Cluster	*Firmicutes*	↑	Grape seed extract (300 mg/kg)	C57BL/6 mice with HFD (60% fat content)	Simultaneous beginning with the experimental diet	7 weeks	[56]
*Coprobacillus*	Genus	*Firmicutes*	↑	Cranberry extract PAs (200 mg/kg)	HFHNCCD mice C57BI/6J (65% lipids, 15% proteins and 20% CH)	13 weeks after HFHNCCD	8 weeks	[67]
*Faecalibacterium praustnitzii*	Species	*Firmicutes*	↑	Carob fruit extract (4 g/kg restructured meat)	HFD mice (50% fat, 1.4% cholesterol and 0.2% cholic acid)	Simultaneous beginning with the experimental diet	8 weeks	[61]
*Lactobacillales*	Order	*Firmicutes*	↑	Blueberry PAs (10% of diet)	HFD mice (45% kcal as fat)	Simultaneous beginning with the experimental diet	8 weeks	[60]
*Lactobacillus*	Genus	*Firmicutes*	↑	Carob fruit extract (4 g/kg restructured meat)	HFD rats (50% fat, 1.4% cholesterol and 0.2% cholic acid)	Simultaneous beginning with the experimental diet	8 weeks	[61]
*Lactobacillus*	Genus	*Firmicutes*	↑	Persimmon tannins (50 or 100 mg/kg)	High-cholesterol Diet Sprague Dawley rats	Simultaneous beginning with the experimental diet	4 weeks	[66]
*Lactobacillus*	Genus	*Firmicutes*	↑	*Pyracantha fortuneana* (0.4% of diet)	HFD rats (Normal diet + 12% fat)	2 weeks after HFD	8 weeks	[68]
*Roseburia*	Genus	*Firmicutes*	↑	Grape seed extract (300 mg/kg)	C57BL/6 mice with HFD (60% fat content)	Simultaneous beginning with the experimental diet	7 weeks	[56]
*Roseburia*	Genus	*Firmicutes*	↑	Cinnamon (2 g/kg) and grape extract (8.2 g/kg)	HFD mice (60% kcal fat diet)	Simultaneous beginning with the experimental diet	8 weeks	[12]

CH: carbohydrates; HFD: high-fat diet, normally referred to high-saturated fat diets; HFHNCCD: high-fat high-non-complex carbohydrates diet; ↑ increase; ↓ decrease.

**Table 2 ijms-24-05369-t002:** Effects of proanthocyanidin consumption on bacteria in which there is ignorance or controversy in relation with metabolic syndrome.

Bacteria	Taxa	Phylum	Effect	Origin and Dose	Experimental Model	PA Introduction	PA Intake Period	Reference
*Actinomycetales*	Order	*Actinobacteria*	↑	*Pyracantha fortuneana* (0.4% of diet)	HFD rats (Normal diet + 12% fat)	2 weeks after HFD	8 weeks	[68]
*Adlercreutzia*	Genus	*Actinobacteria*	↑	Blueberry powder PAs (160 mg)	HFHNCCD mice (65% fat + 18% sucrose)	Simultaneous beginning with the experimental diet	12 weeks	[62]
*Adlercreutzia equolifaciens*	Species	*Actinobacteria*	↑	Wild blueberry PAs (200 mg/kg)	HFHNCCD mice (65% lipids, 15% proteins and 20% CH)	Simultaneous beginning with the experimental diet	8 weeks	[63]
*Bacteroides*	Genus	*Bacteroidetes*	↑	Carob fruit extract (4 g/kg restructured meat)	HFD rats (50% fat, 1.4% cholesterol and 0.2% cholic acid)	Simultaneous beginning with the experimental diet	8 weeks	[61]
*Bacteroides*	Genus	*Bacteroidetes*	↓	Grape seed extract (300 mg/Kg/day)	C57BL/6 mice with HFD (60% fat content)	Simultaneous beginning with the experimental diet	7 weeks	[56]
*Bacteroides*	Genus	*Bacteroidetes*	↓	*Pyracantha fortuneana* (0.4% of diet)	HFD rats (Normal die t+ 12% fat)	2 weeks after HFD	8 weeks	[68]
*Barnesiella*	Genus	*Bacteroidetes*	↑	Cranberry extract PAs (200 mg/kg)	HFHNCCD mice C57BI/6J (65% lipids, 15% proteins and 20% CH)	13 weeks after HFHNCCD	8 weeks	[67]
*Bilophila*	Genus	*Proteobacteria*	↓	B2 procyanidin (0.2%)	C57BL/6 mice with HFD (34% fat diet)	Simultaneous beginning with the experimental diet	8 weeks	[64]
*Blautia*	Genus	*Firmicutes*	↑	B2 procyanidin (0.2%)	C57BL/6 mice with HFD (34% fat diet)	Simultaneous beginning with the experimental diet	8 weeks	[64]
*Blautia*	Genus	*Firmicutes*	↑	Procyanidin (100 mg/kg)	C57BL/6 mice with HFD (60% kcal from fat)	Simultaneous beginning with the experimental diet	12 weeks	[65]
*Blautia*	Genus	*Firmicutes*	↑	Grape seed extract (300 mg/Kg/day)	C57BL/6 mice with HFD (60% fat content)	Simultaneous beginning with the experimental diet	7 weeks	[56]
*Blautia*	Genus	*Firmicutes*	↑	*Pyracantha fortuneana* (0.4% of diet)	HFD rats (Normal die t+ 12% fat)	2 weeks after HFD	8 weeks	[68]
*Blautia coccoides - Eubacterium rectale*	Group	*Firmicutes*	↓	Carob fruit extract (4 g/kg restructured meat)	HFD rats (50% fat, 1.4% cholesterol and 0.2% cholic acid)	Simultaneous beginning with the experimental diet	8 weeks	[61]
*Blautia coccoides - Eubacterium rectale*	Group	*Firmicutes*	↓	Carob fruit extract (4 g/kg restructured meat)	HFD rats (50% fat, 1.4% cholesterol and 0.2% cholic acid)	3 weeks after HFD	5 weeks	[61]
*Clostridium leptum*	Group	*Firmicutes*	↓	Carob fruit extract (4 g/kg restructured meat)	HFD rats (50% fat, 1.4% cholesterol and 0.2% cholic acid)	Simultaneous beginning with the experimental diet	8 weeks	[61]
*Lachnoclostridium*	Genus	*Firmicutes*	↓	B2 procyanidin (0.2%)	C57BL/6 mice with HFD (34% fat diet)	Simultaneous beginning with the experimental diet	8 weeks	[64]
*Lachnospiraceae*	Family	*Firmicutes*	↓	Procyanidin (100 mg/kg)	C57BL/6 mice with HFD (60% kcal from fat)	Simultaneous beginning with the experimental diet	12 weeks	[65]
*Muribaculum intestinale*	Species	*Bacteroidetes*	↑	Blueberry powder PAs (160 mg)	HFHNCCD mice (65% fat + 18% sucrose)	Simultaneous beginning with the experimental diet	12 weeks	[62]
*Oscillospira*	Genus	*Firmicutes*	↓	*Pyracantha fortuneana* (0.4% of diet)	HFD rats (Normal die t+ 12% fat)	2 weeks after HFD	8 weeks	[68]
*Proteobacteria*	Phylum	*Proteobacteria*	↑	Grape seed extract (300 mg/Kg/day)	C57BL/6 mice with HFD (60% fat content)	Simultaneous beginning with the experimental diet	7 weeks	[56]
*Proteobacteria*	Phylum	*Proteobacteria*	↑	Blueberry PAs (10% of diet)	HFD mice (45% kcal as fat)	Simultaneous beginning with the experimental diet	8 weeks	[60]
*Proteobacteria*	Phylum	*Proteobacteria*	↑	B2 procyanidin (0.2%)	C57BL/6 mice with HFD (34% fat diet)	Simultaneous beginning with the experimental diet	8 weeks	[64]
*Proteus*	Genus	*Proteobacteria*	↓	B2 procyanidin (0.2%)	C57BL/6 mice with HFD (34% fat diet)	Simultaneous beginning with the experimental diet	8 weeks	[64]
*Ruminococcus*	Genus	*Firmicutes*	↓	*Pyracantha fortuneana* (0.4% of diet)	HFD rats (Normal die t+ 12% fat)	2 weeks after HFD	8 weeks	[68]
*Ruminococcaceae*	Family	*Firmicutes*	↓	Procyanidin (100 mg/kg)	C57BL/6 mice with HFD (60% kcal from fat)	Simultaneous beginning with the experimental diet	12 weeks	[65]
*Turicibacterales*	Order	*Firmicutes*	↑	*Pyracantha fortuneana* (0.4% of diet)	HFD rats (Normal diet + 12% fat)	2 weeks after HFD	8 weeks	[68]

CH: carbohydrates; HFD: high-fat diet, normally referred to high-saturated fat diets; HFHNCCD: high-fat high-non-complex carbohydrates diet; ↑ increase; ↓ decrease. White color responds to bacteria with a not well-established relationship with MS. Gray color responds to bacteria with a controversial relationship with MS.

**Table 3 ijms-24-05369-t003:** Effects of proanthocyanidin consumption on bacteria considered deleterious in metabolic syndrome.

Bacteria	Taxa	Phylum	Effect	Origin and Dose	Experimental Model	PAs Introduction	PAs Intake Period	Reference
*Desulfovibrio*	Genus	*Proteobacteria*	↓	Procyanidin (100 mg/kg)	C57BL/6 mice with HFD (60% kcal from fat)	Simultaneous beginning with the experimental diet	12 weeks	[65]
*Desulfovibrionaceae*	Family	*Proteobacteria*	↓	Cinnamon (2 g/kg) and grape extract (8.2 g/kg)	HFD mice (60% kcal fat diet)	Simultaneous beginning with the experimental diet	8 weeks	[12]
*Desulfovibrionaceae*	Family	*Proteobacteria*	↓	*Pyracantha fortuneana* (0.4% of diet)	HFD rats (Normal diet + 12% fat)	2 weeks after HFD	8 weeks	[68]
*Enterobacteriaceae*	Family	*Proteobacteria*	↓	Carob fruit extract (4 g/kg restructured meat)	HFD rats (50% fat, 1.4% cholesterol and 0.2% cholic acid)	3 weeks after HFD	5 weeks	[61]
*Enterococcus*	Genus	*Firmicutes*	↓	Persimmon tannins (50 or 100 mg/kg)	High-cholesterol Diet Sprague-Dawley rats	Simultaneous beginning with the experimental diet	4 weeks	[66]
*Enterococcus*	Genus	*Firmicutes*	↓	Carob fruit extract (4 g/kg restructured meat)	HFD rats (50% fat, 1.4% cholesterol and 0.2% cholic acid)	3 weeks after HFD	5 weeks	[61]
*Lactococcus*	Genus	*Firmicutes*	↓	Grape seed extract (300 mg/Kg/day)	C57BL/6 mice with HFD (60% fat content HFD)	Simultaneous beginning with the experimental diet	7 weeks	[56]
*Lactococcus*	Genus	*Firmicutes*	↓	Cinnamon (2 g/kg) and grape extract (8.2 g/kg)	HFD mice (60% kcal fat diet)	Simultaneous beginning with the experimental diet	8 weeks	[12]
*Firmicutes*	Phylum	*Firmicutes*	↓	Blueberry PAs (10% of diet)	HFD mice (45% kcal as fat)	Simultaneous beginning with the experimental diet	8 weeks	[60]
*Firmicutes*	Phylum	*Firmicutes*	↓	Grape seed extract (300 mg/Kg/day)	C57BL/6 mice with HFD (60% fat content)	Simultaneous beginning with the experimental diet	7 weeks	[56]
*Firmicutes*	Phylum	*Firmicutes*	↓	B2 procyanidin (0.2%)	C57BL/6 mice with HFD (34% fat diet)	Simultaneous beginning with the experimental diet	8 weeks	[64]
*Firmicutes*	Phylum	*Firmicutes*	↓	Persimmon tannins (50 or 100 mg/kg)	High-cholesterol Diet Sprague-Dawley rats	Simultaneous beginning with the experimental diet	4 weeks	[66]

HFD: high-fat diet, normally referred to high-saturated fat diets; HFHNCCD: high-fat high-non-complex carbohydrates diet; ↑ increase; ↓ decrease.

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
