# Peer review of "Proanthocyanidins: Impact on Gut Microbiota and Intestinal Action Mechanisms in the Prevention and Treatment of Metabolic Syndrome"

_ijms, 2023, doi:10.3390/ijms24065369_

Round 1
Reviewer 1 Report
This review offers a complete summary of the works that have been made in rodents with metabolic syndrome (MS) taking proanthocyanidins (PAs) and their impact on gut microbiota, which is of great value. In addition, a system to facilitate comparison among studies is provided, which is also very relevant to the field. However, this work lacks contextualization in some important aspects: it lacks details on how intestinal dysbiosis is both an aggravating factor and consequence of MS, and how the modulation of gut microbiota is positive for treating MS, which is stated to be the focus of this work. Besides, this work is unbalanced in regards to introducing the right terms for the later explanation of the positive effects PAs, such as the immune homeostasis of the gastrointestinal tract: it is bearly explained in the first sections but is again mentioned later when explaining the benefits of PAs as immunomodulators. Thus, this works needs some reorganization.
Comments:
- Title: the focus on the effect of PAs on gut microbiota should be mentioned.
- Abstract: as before mentioned, there should be a shift of focus toward the effect of PAs on gut microbiota.
- Part 2.1: Both high and low F/B ratios are a marker of dysbiosis. Increased F/B is associated with obesity and decreased F/B is associated with inflammatory bowel disease (IBD).
- Part 2.2: the amount of detail in the imbalance of immune homeostasis in the intestine during MS should be increased in order to properly introduce the effects of PAs later in the manuscript. The role of cytokine regulation of tight junctions should be mentioned.
- Part 2.3: although the figure supporting this section contains the most important aspects of the connection between intestinal dysbiosis and MS development, the text is incomplete. As before mentioned, in this section there should be an explanation of MS development and its relation with intestinal dysbiosis.
- Part 3: although the focus of this work is on PAs to manage MS, the other approaches used to treat MS should be mentioned to provide context (pharmacological, surgical...).
- Part 5.1: here, bacteria are classified as good, controversial or deleterious for MS. Nevertheless, only a few comments address why are some of these bacteria relevant in the context of MS: such as low firmicutes/Bacteroidetes ratio is associated with improved glucose tolerance. The measured outcome in terms of MS progression/treatment with PAs should be included in the text or tables (for example: in table 1, besides which beneficial bacteria strain was increased, how did the treatment with PAs impact the MS? you could include a summary of the MS-related markers measured in the listed studies: improved glucose tolerance/reduced plasma NEFA...).
- You could mention that not only diet is a cause of intestinal dysbiosis in MS, but also in other inflammatory conditions such as IBS/Crohn's and that the therapeutic properties of PAs could be beneficial in these other pathologies.
Author Response
Please see the attachment for greater details:
This review offers a complete summary of the works that have been made in rodents
with metabolic syndrome (MS) taking proanthocyanidins (PAs) and their impact on gut
microbiota, which is of great value. In addition, a system to facilitate comparison among
studies is provided, which is also very relevant to the field. However, this work lacks
contextualization in some important aspects: it lacks details on how intestinal dysbiosis
is both an aggravating factor and consequence of MS, and how the modulation of gut
microbiota is positive for treating MS, which is stated to be the focus of this work.
Besides, this work is unbalanced in regards to introducing the right terms for the later
explanation of the positive effects PAs, such as the immune homeostasis of the
gastrointestinal tract: it is bearly explained in the first sections but is again mentioned
later when explaining the benefits of PAs as immunomodulators. Thus, this works
needs some reorganization.
Thank you very much for the good evaluation of our manuscript and for
considering some points relevant to the field. We are delighted to consider your
proposals and improve the manuscript quality.
The reviewer has some concerns as follows:
1. Title: the focus on the effect of PAs on gut microbiota should be mentioned.
Thank you for your suggestion. Actually, the effects on gut microbiota are a
crucial aspect. We have changed the title as follows: “Proanthocyanidins: Impact
on Gut Microbiota and Intestinal Action Mechanisms in the Prevention and
Treatment of Metabolic Syndrome”.
2. Abstract: as before mentioned, there should be a shift of focus toward the effect of
PAs on gut microbiota.
We appreciate the emphasis on this question as we believe that it allows the
reader to get from the beginning a better idea of what is developed along the
review. Therefore, we have added the following sentence at the end of the
abstract (line 29): “Special emphasis has been made on the impact of gut
microbiota, providing a system to facilitate comparison between the studies”.
3. Part 2.1: Both high and low F/B ratios are a marker of dysbiosis. Increased F/B is
associated with obesity and decreased F/B is associated with inflammatory bowel
disease (IBD).
Thank you for your appointment. Re-reading the sentence we agree that it is
quite confusing. We wanted to highlight in this case the high F/B ratio linked to
obesity, as we have focused on HFD-associated MS. To clarify the sentence we
have modified it, as follows (lines 116-119): “Therefore, an increased
Firmicutes/Bacteroidetes (F/B) ratio is an accepted marker of obesity- associated
dysbiosis [13]. This dysbiosis is considered to play a key role in the origin and
development of MS, representing the main environmental factor contributing to
MS [14]”.
4. Part 2.2: the amount of detail in the imbalance of immune homeostasis in the
intestine during MS should be increased in order to properly introduce the effects of
PAs later in the manuscript. The role of cytokine regulation of tight junctions should be
mentioned.
We acknowledge your consideration. We decided to shorten the final version of
the manuscript, before sending it to the journal, by removing part of the
introduction. Now, after your comment, we realize that this was not the best idea.
We have added this information in the line 163 of this second version, as follows:
“In this context, cells of gastrointestinal innate immunity (involving epithelial
cells and antigen-presenting cells found in the lamina propria) that exhibit Toll-
type receptors (TLRs) on their membrane, recognize lipopolysaccharide (LPS)
from Gram-negative bacteria and activate inflammatory pathways—primarily
those that promote the activation of mitogen-activated protein kinases (MAPK)
and nuclear factor kappa-light-chain-enhancer of activated B cells (NF-κB)
pathways. This event leads to overexpression of TLRs, and release of
inflammatory mediators, and transcription of inducible enzymes related with the
inflammatory and oxidative process. Moreover, MAPK activation favors the
blockage in the MLCK transcription, finally promoting the disorganization of TJs
and amplifying the pathological barrier hyperpermeability. In this sense, it has
been demonstrated that proinflammatory mediators directly induce TJs
disruption, which dramatically increases intestinal permeability and facilitates
bacterial translocation [9]. Oxidative stress and inflammatory cytokines
overactivate inflammatory pathways in a complex feedback loop, which act by
amplifying barrier dysfunction and perpetuating the inflammatory state
[26,28,29]. Consequently, dysbiosis and intestinal barrier dysfunction is one of
the most important factors connecting diet and intestinal dysbiosis to the
development of systemic metabolic disorders [9]...
5. Part 2.3: although the figure supporting this section contains the most important
aspects of the connection between intestinal dysbiosis and MS development, the text is
incomplete. As before mentioned, in this section there should be an explanation of MS
development and its relation with intestinal dysbiosis.
Thank you for your statement. After reading your explanation, we have realized
that placing Figure 1 at the end of part 2.3. may be confusing. In this second
version of the manuscript, Figure 1 has been moved to the beginning of section
2 -Loss of intestinal homeostasis as an etiological factor of metabolic syndrome
and the importance of consuming a high-fat diet- as it summarizes the
information of subsections 2.1., 2.2., and 2.3.
About your second comment, the connection between gut dysbiosis and the
development of MS is covered in the whole point 2, specifically in subsection
2.1.- Intestinal Dysbiosis and Alterations in the SCFAs Profile Promoted by High-
Fat Diets and Associated with Metabolic Syndrome. Regarding the development
of MS, and agreeing that dysbiosis is a critical event, we have highlighted the
role of gut barrier dysfunction instead of focusing just on dysbiosis.
Nonetheless, and taking into account your words, we have expanded the
information with a further description of the systemic metabolic disturbances
and insulin resistance that trigger the onset of MS, adding also a recent
reference about this topic (Thomas et al., 2022). Thus, the following sentence can
be found in lines 190-195 of this second version of the manuscript: “It favors
bacterial translocation in a positive feedback loop, causing systemic
disturbances increases body weight, induces insulin resistance leading to
hyperinsulinemia and hyperglycemia, and activates systemic inflammatory
pathways [10,28,32]. Hence, intestinal dysbiosis and barrier dysfunction
ultimately promote insulin resistance, obesity, inflammation, dyslipidemia, and
even hypertension, all of them important features of MS [33].”
6. Part 3: although the focus of this work is on PAs to manage MS, the other
approaches used to treat MS should be mentioned to provide context
(pharmacological, surgical...).
Thank you for your comments. We agree with you about this consideration, as it
makes the section more complete and provides adequate context for the reader.
Thus, we have extended Section 3- Metabolic syndrome management - including
other approaches to MS treatment (lines 200-233) “MS comprises some
comorbidities that are better to treat in order to avoid further health
consequences such as T2DM and cardiovascular disease. Changes in lifestyles
and pharmacological treatment are the two main strategies contemplated in the
MS management [34]. Besides, bioactive compounds and nutraceuticals can be a
first step for treating short clinical crows before resorting to the use of drugs, as
they are able to exert innumerable potential biological actions.
Because HFD consumption is an inductor of MS, dietary interventions and
targeted nutritional therapies could provide great promise for its prevention and
treatment. Adopting a healthy lifestyle is already the cornerstone of MS
management. Among the dietary interventions to reduce the incidence of MS,
increased intake of complex carbohydrates and lean proteins is recommended,
as well as limit the intake of saturated fat. In addition, increasing the
consumption of fiber, even in the framework of a HFD, could alleviate many of its
negative effects, especially those related to the intestinal barrier, where fiber
improves the composition of the microbiome.
Regarding pharmacological strategy, a poly-pharmacological therapy is often
required [35]. The approach includes drugs that, in addition to their specific
indications (hypoglycemic, hypolipidemic, hypotensive, etc.), combine their
effect increasing insulin sensitivity in peripheral tissues. Hence,
antihyperglycemic agents such as pioglitazone or dipeptidyl peptidase-4
inhibitors, sitagliptin, have been shown to be beneficial in MS patients. More
recently, liraglutide, a GLP-1 receptor agonist, has been investigated and used in
clinical practice for its direct anti-atherosclerotic action. Likewise, statins have
been studied for their anti-inflammatory effect by reducing blood levels of high-
sensitivity C-reactive protein, and for their beneficial antithrombotic role in MS.
On the other hand, currently, it is being widely investigated the role of
phytochemicals, bioactive compounds and nutraceuticals (plant extracts, spices,
herbs and essential oils) in the treatment of MS [36]. In need of further research,
it has been obtained positive results that make them a promising alternative for
the development of new therapies. Among bioactive compounds, flavonoids
stand out because of the large number of studies investigating their efficacy
against MS. For example, the possible preventive role of soy isoflavones in
metabolic syndrome-induced cardiovascular disease is worth mentioning.
Another flavonoid of interest in ameliorating the signs of MS is quercetin due to
its antihypertensive, antihyperlipidaemic, antihyperglycaemic and other
properties [37]. For more information, we recommend consulting the review by
Gouveia et al. [38].”
7. Part 5.1: here, bacteria are classified as good, controversial or deleterious for MS.
Nevertheless, only a few comments address why are some of these bacteria relevant
in the context of MS: such as low firmicutes/Bacteroidetes ratio is associated with
improved glucose tolerance. The measured outcome in terms of MS
progression/treatment with PAs should be included in the text or tables (for example: in
table 1, besides which beneficial bacteria strain was increased, how did the treatment
with PAs impact the MS? you could include a summary of the MS-related markers
measured in the listed studies: improved glucose tolerance/reduced plasma NEFA...).
We appreciate your comment, and we agree about the importance of this point.
Although we consider that relating changes in specific bacteria with metabolic
consequences would be the best option; it is really difficult to attribute a
beneficial effect just to the modulation of one bacteria taxa. Hence, some of the
revised articles (not all) give information about MS outcomes and GM
homeostasis restoration, but without attributing them to specific bacteria taxa.
Finally, it is important to consider that we have organized the tables based on
bacteria-specific changes, instead of articles that analyze this matter. For this
reason, including the requested information in the tables is complicated,
because the same study is repeated in different places of the tables, according
to the number of analyzed bacteria. So, information about outcomes in MS
progression would also be repeated every time and could be confusing for the
reader.
8. You could mention that not only diet is a cause of intestinal dysbiosis in MS, but also
in other inflammatory conditions such as IBS/Crohn's and that the therapeutic
properties of PAs could be beneficial in these other pathologies..
Thank you for your suggestion. Diet is involved in the origin of many
pathologies, intestinal or not, and PAs could be interesting for preventing or
treating most of them. However, and keeping in mind the amount of information
published about PAs, we have tried to focus on the key aspects to achieve the
objective of this review. Definitely, the study of these prevalent pathologies and
their relationship with PAs is an interesting topic that we will consider for a
future review.

Reviewer 2 Report
The manuscript entitled “Proanthocyanidins: Intestinal Action Mechanisms in the Prevention and Treatment of Metabolic Syndrome”, and authored by Rocío Redondo-Castillejo and colleagues, deals with the current evidence on the mechanisms involved in the preventive or curative role of PAs-rich plant extracts to maintain or restore intestinal homeostasis in HFD-induced MS.
The manuscript is really well written, summarizing attractive information obtained from various bibliographic sources. The structure of the sections and subsections follows a very precise and rational logic. Conclusively, I think it would be a shame not to accept this manuscript as a potential publication in IJMS.
However, before the manuscript can be judged suitable as a publication, some small changes are recommended to be made.
· Some keywords should be changed. The utility of these terms is to facilitate the search of the article using common scientific search engines (PubMed, GoogleScholar, Scopus, etc.), which rely on the terms contained in title, abstract, and keywords. Consequently, using terms that are already in these sections as keywords is inappropriate. I strongly suggest that the repetitive keywords be changed before re-submission.
· At the end of the introduction (Chapter 1), authors should include literature references in which MS have been treated using botanicals, phytochemicals, or bioactive plant secondary compounds. This could serve to motivate their investigation. For example: (i) In 10.3390/ijms22052664, docking studies along with biochemical, cellular, and biomolecular studies probed the effects of triterpenes from protium gum resin to low cholesterol levels; (ii) In 10.3389/fimmu.2013.00132, polyphenols from green tea improved antioxidants levels and attenuated severity of colitis analogous to sulfasalazine in colitis-induced mice; In 10.1016/j.jnutbio.2012.11.008, the protective effect of polyphenols extracted from dietary extra virgin olive oil showed inflammatory activity against to chronic colitis in mice model system; etc..
· Figure 2 is superfluous, and could be removed, or moved as supplementary. the affiliations section should include each author's email, along with their acronym. This acronym should be the same one used in the contributions section.
· The abstract should be implemented. It contains far too much information related to the current state of the art, but little instead related to the structure of the manuscript and the authors' focuses. Accordingly, it should be implemented.
· Chapter 3 should be slightly implemented. In particular, this section could be useful to link the previous chapters to Chapter 4, which currently seems to be an end in itself. For example, the authors could report that before resorting to the use of drugs, short clinical pictures can be treated as a first step with botanicals or phytopharmaceuticals. These particular formulations, although not considered drugs in their own right, contain bioactive components that can exert innumerable potential biological actions. Among these, PAs-based botanicals have been shown .....
· The data presented in Tables 1-3 lend themselves well to a forest plot. Do the authors happen to have the ability to analyze these data to create forest plots? If the authors have no other software, a free software for creating forest plots could be RevManager
Author Response
Please see the attachment for greater details:
The manuscript entitled “Proanthocyanidins: Intestinal Action Mechanisms in the
Prevention and Treatment of Metabolic Syndrome”, and authored by Rocío Redondo-
Castillejo and colleagues, deals with the current evidence on the mechanisms involved
in the preventive or curative role of PAs-rich plant extracts to maintain or restore
intestinal homeostasis in HFD-induced MS.
The manuscript is really well written, summarizing attractive information obtained from
various bibliographic sources. The structure of the sections and subsections follows a
very precise and rational logic. Conclusively, I think it would be a shame not to accept
this manuscript as a potential publication in IJMS.
However, before the manuscript can be judged suitable as a publication, some small
changes are recommended to be made.:
Thank you very much for considering our manuscript as a potential publication
in IJMS, and for helping us to improve it. We appreciate your detailed evaluation
and accept your suggestions/corrections to improve the reading,
comprehension, and quality of our manuscript.
Some keywords should be changed. The utility of these terms is to facilitate the search
of the article using common scientific search engines (PubMed, GoogleScholar,
Scopus, etc.), which rely on the terms contained in title, abstract, and keywords.
Consequently, using terms that are already in these sections as keywords is
inappropriate. I strongly suggest that the repetitive keywords be changed before re-
submission.
We are very grateful for this comment because this is something we were
completely unaware of. In view of the importance of the information, and
following your recommendations, we have incorporated some changes to the
keywords. For example, instead of gut microbiota we have included words such
as dysbiosis and bioactive compounds. We have also added metabolic
syndrome management instead of metabolic syndrome
At the end of the introduction (Chapter 1), authors should include literature references
in which MS have been treated using botanicals, phytochemicals, or bioactive plant
secondary compounds. This could serve to motivate their investigation. For example:
(i) In 10.3390/ijms22052664, docking studies along with biochemical, cellular, and
biomolecular studies probed the effects of triterpenes from protium gum resin to low
cholesterol levels; (ii) In 10.3389/fimmu.2013.00132, polyphenols from green tea
improved antioxidants levels and attenuated severity of colitis analogous to
sulfasalazine in colitis-induced mice; In 10.1016/j.jnutbio.2012.11.008, the protective
effect of polyphenols extracted from dietary extra virgin olive oil showed inflammatory
activity against to chronic colitis in mice model system; etc..
We acknowledge your appreciation. We agree that the use of natural compounds
to treat MS and other intestinal diseases as chronic colitis is increasing. Despite
this fact, new investigations demonstrating their mechanisms and efficacy are
also useful. Following your pertinent comments, we have specified the
alternative of using these products in MS treatment, and we have also added a
new literature reference in the second version of the manuscript. They can be
found in chapter 3 “metabolic syndrome management” instead of chapter 1
about general aspects of MS.
Figure 2 is superfluous, and could be removed, or moved as supplementary. The
affiliations section should include each author's email, along with their acronym. This
acronym should be the same one used in the contributions section.
Thank you for your comments. In accordance with your first recommendation,
we have decided to delete figure 2 as superfluous. At the same time, we have
subsequently corrected the enumeration of the rest of the figures presented in
the manuscript. Thus, figure 2 becomes the one presenting the main
subcategories of flavonoids. Figure 3 is now the one with types A and B of
proanthocyanidins. Figure 6 becomes figure 4, as there was a mistake in the first
version of the manuscript; and figure 5 is numbered as such.
Regarding the second comment concerning the authors ́ data, we appreciate the
guidance for the improvement of the paper. However, we apologize for not
applying this recommendation, as we are not sure that it is necessary according
to the journal's instructions. Nonetheless, if the editor considers it appropriate,
and provides us with the details to do so, we will be glad to complete this
information.
The abstract should be implemented. It contains far too much information related to the
current state of the art, but little instead related to the structure of the manuscript and
the authors' focuses. Accordingly, it should be implemented.
We appreciate your consideration. To help the reader comprehension, we have
deeply modified the abstract in this second version of the manuscript,
highlighting our focuses and conclusions.
Chapter 3 should be slightly implemented. In particular, this section could be useful to
link the previous chapters to Chapter 4, which currently seems to be an end in itself.
For example, the authors could report that before resorting to the use of drugs, short
clinical pictures can be treated as a first step with botanicals or phytopharmaceuticals.
These particular formulations, although not considered drugs in their own right, contain
bioactive components that can exert innumerable potential biological actions. Among
these, PAs-based botanicals have been shown...
Thank you for your consideration. It should be noted that Referee 1 has also
made a comment to us in this regard. We have considered both, and thus
chapter 3 has been completed taking into account pharmacological aspects and
bioactive components as part of the treatment of MS, as follows (lines 200 and
233): “MS comprises some comorbidities that are better to treat in order to avoid
further health consequences such as T2DM and cardiovascular disease.
Changes in lifestyles and pharmacological treatment are the two main strategies
contemplated in the MS management [34]. Besides, bioactive compounds and
nutraceuticals can be a first step for treating short clinical crows before
resorting to the use of drugs, as they are able to exert innumerable potential
biological actions.
Because HFD consumption is an inductor of MS, dietary interventions and
targeted nutritional therapies could provide great promise for its prevention and
treatment. Adopting a healthy lifestyle is already the cornerstone of MS
management. Among the dietary interventions to reduce the incidence of MS,
increased intake of complex carbohydrates and lean proteins is recommended,
as well as limit the intake of saturated fat. In addition, increasing the
consumption of fiber, even in the framework of a HFD, could alleviate many of its
negative effects, especially those related to the intestinal barrier, where fiber
improves the composition of the microbiome.
Regarding pharmacological strategy, a poly-pharmacological therapy is often
required [35]. The approach includes drugs that, in addition to their specific
indications (hypoglycemic, hypolipidemic, hypotensive, etc.), combine their
effect increasing insulin sensitivity in peripheral tissues. Hence,
antihyperglycemic agents such as pioglitazone or dipeptidyl peptidase-4
inhibitors, sitagliptin, have been shown to be beneficial in MS patients. More
recently, liraglutide, a GLP-1 receptor agonist, has been investigated and used in
clinical practice for its direct anti-atherosclerotic action. Likewise, statins have
been studied for their anti-inflammatory effect by reducing blood levels of high-
sensitivity C-reactive protein, and for their beneficial antithrombotic role in MS.
On the other hand, currently, it is being widely investigated the role of
phytochemicals, bioactive compounds and nutraceuticals (plant extracts, spices,
herbs and essential oils) in the treatment of MS [36]. In need of further research,
it has been obtained positive results that make them a promising alternative for
the development of new therapies. Among bioactive compounds, flavonoids
stand out because of the large number of studies investigating their efficacy
against MS. For example, the possible preventive role of soy isoflavones in
metabolic syndrome-induced cardiovascular disease is worth mentioning.
Another flavonoid of interest in ameliorating the signs of MS is quercetin due to
its antihypertensive, antihyperlipidaemic, antihyperglycaemic and other
properties [37]. For more information, we recommend consulting the review by
Gouveia et al. [38].” We believe that the manuscript has been improved by the
changes introduced.
The data presented in Tables 1-3 lend themselves well to a forest plot. Do the authors
happen to have the ability to analyze these data to create forest plots? If the authors
have no other software, a free software for creating forest plots could be RevManager
We are very grateful for your suggestion. We find the forest plot very interesting,
but we were not aware of the possibility of a free program. We have been testing
and considering to present the data using this tool, but unfortunately the
deadline to give an answer is short, and the first time you use the program is not
easy and demands more time.

Reviewer 3 Report
I have now reviewed the article: Proanthocyanidins: Intestinal Action Mechanisms in the Prevention and Treatment of Metabolic Syndrome.
The authors have done great work in collecting and classifying the data on the relationship of PAs and Intestinal action mechanisms. However, the sections on the Results Obtained from Rodent Experiments and Clinical trials are disorganized and confusing. I have the following points to improve this valuable review:
1. The authors have discussed a good introduction on MS and causes. However, I was surprised that they did not mention the role of Stress and its hormone, Cortisol as a major contributor of high-grade inflammation. Certainly, Cushing Syndromes are considered another component of MS.
2. I would liked a small section on the role of other flavonoids on MS and Intestinal action mechanisms in comparisons with PAs.
3. Section 5.1 Rodent Experiments is far too long without breaks and not easy to read and follow. I suggest it can be sub-divided into few sub-sections.
4. I was hoping that the authors will discuss briefly the results and the data mention in Tables 1, 2 and 3, especially there bacteria that is beneficial, beneficial or (ignorance or controversy) in relation with metabolic syndrome. Highlighting some of the studies is not enough.
5. Section 5.2 Results obtained in clinical trials is very weak and not interesting. Perhaps the authors can mention also some of the Pre-clinical studies which might add more data to their review.
6. Sections of Absorption, metabolism, and excretion of proanthocyanidins. Biotransformation by the microbiota and intestinal epithelial barrier integrity by dietary proanthocyanidins are well written and nice to read.
7. some minor points:
a) The authors mentioned PA extracts in the aim of the review. Then they mention real diet as fruits and vegetables. Please choose.
b) List of abbreviations are needed at the end of the review.
c) The authors proposes several criteria in some sections. Are these directed to researchers, policy makers or research bodies? Have they passed such recommendations to their research authorities?
Author Response
Please see the attachment for more details:
I have now reviewed the article: Proanthocyanidins: Intestinal Action Mechanisms in
the Prevention and Treatment of Metabolic Syndrome.
The authors have done great work in collecting and classifying the data on the
relationship of PAs and Intestinal action mechanisms. However, the sections on the
Results Obtained from Rodent Experiments and Clinical trials are disorganized and
confusing. I have the following points to improve this valuable review:
We acknowledge you appreciate our work collecting relevant data. We also thank
you very much for all your comments that we have kept in mind to improve the
comprehension and quality of data discussion.
1. The authors have discussed a good introduction on MS and causes. However, I was
surprised that they did not mention the role of Stress and its hormone, Cortisol as a
major contributor of high-grade inflammation. Certainly, Cushing Syndromes are
considered another component of MS.
Thank you for your consideration of the introduction. We also consider the role
of cortisol in MS interesting and relevant. However, we have focused on the loss
of intestinal homeostasis due to HFD intake as an important and early driver of
MS. There are many other systemic alterations that we have not mentioned in the
introduction just because we did not discuss anything about them in the rest of
the review. We have looked at the bibliography, and we have enjoyed these
papers (https://academic.oup.com/jcem/article/94/8/2692/2596309?login=true and
https://www.ncbi.nlm.nih.gov/pmc/articles/PMC8978123/). It is discussed that
cortisol impairs gut barrier via modifying TJs. We have kept in mind that we
wanted to analyze the role of PAs in protecting healthy microbiota and gut
barrier integrity from the harmful effect of consuming a HFD. Preclinical models
and clinical assays have been carefully chosen in this sense. The ability of PAs
preventing cortisol increased and the evaluation of the systemic and intestinal
consequences of it is an interesting topic that we will consider for a future
review.
2. I would like a small section on the role of other flavonoids on MS and Intestinal
action mechanisms in comparisons with PAs.
Thank you for your consideration. Following your comment, and those of the
other two reviewers concerning the use of bioactive compounds in the treatment
of MS, we have added some information about flavonoids in section 3. It can be
found in lines 223-233: “On the other hand, currently, it is being widely
investigated the role of phytochemicals, bioactive compounds and
nutraceuticals (plant extracts, spices, herbs and essential oils) in the treatment
of MS [36]. In need of further research, it has been obtained positive results that
make them a promising alternative for the development of new therapies. Among
bioactive compounds, flavonoids stand out because of the large number of
studies investigating their efficacy against MS. For example, the possible
preventive role of soy isoflavones in metabolic syndrome-induced
cardiovascular disease is worth mentioning. Another flavonoid of interest in
ameliorating the signs of MS is quercetin due to its antihypertensive,
antihyperlipidaemic, antihyperglycaemic and other properties [37]. For more
information, we recommend consulting the review by Gouveia et al. [38].”
3. Section 5.1 Rodent Experiments is far too long without breaks and not easy to read
and follow. I suggest it can be sub-divided into few sub-sections.
Thank you for your appreciation, it has motivated us to reconsider the structure
of section 5.1. and to subdivide it into different sub-sections (5.1.1 to 5.1.4), as
you propose, helping to follow the paper more easily.
4. I was hoping that the authors will discuss briefly the results and the data mention in
Tables 1, 2 and 3, especially there bacteria that is beneficial, beneficial or (ignorance or
controversy) in relation with metabolic syndrome. Highlighting some of the studies is
not enough.
Excuse us, we are very sorry for not having a clear understanding of this
comment. We believe that we properly discussed the studies that we have
collected in tables 1 to 3, and also we have compared them and extracted
conclusions. Perhaps, now that it is subdivided, as you kindly requested, it is
easy to find the interpretations of results. Nonetheless, if you please provide us
more specific details to do so, we will be glad to complete this information.
5. Section 5.2 Results obtained in clinical trials is very weak and not interesting.
Perhaps the authors can mention also some of the Pre-clinical studies which might add
more data to their review.
We appreciate your suggestion. We believe that precisely the weakness of the
results obtained through clinical studies highlights the importance of the
distinction between prevention and treatment in pre-clinical experiments in order
to properly design clinical trials. The inconsistency of the results obtained in
clinical trials is probably due to the lack of publication of negative results. We
hope, from this point of view, you will find this section more interesting to read.
6. Sections of Absorption, metabolism, and excretion of proanthocyanidins.
Biotransformation by the microbiota and intestinal epithelial barrier integrity by dietary
proanthocyanidins are well written and nice to read.
We appreciate your comment on this section of the paper and we are glad that
you found it enjoyable to read.
7. Some minor points:
a) The authors mentioned PA extracts in the aim of the review. Then they mention real
diet as fruits and vegetables. Please choose.
Thank you for this observation. We have proceeded to remove "PA extract" and
“fruits and vegetables” because in the introduction we want to reflect a broader
aspect and we do not want to make the reader fall into contradictions.
b) List of abbreviations are needed at the end of the review.
We acknowledge your appreciation. As indicated in the journal guidelines, we
have defined acronyms/abbreviations/initialisms the first time they appear in
each of the three sections: the abstract; the main text; the first figure or table. To
this end, we have added the acronym/abbreviation/initialism in parentheses after
the written form, when they are defined for the first time. However, if the Editor
thinks it is necessary, we will be delighted to generate a list of abbreviations.
c) The authors propose several criteria in some sections. Are these directed to
researchers, policy makers or research bodies? Have they passed such
recommendations to their research authorities? I think they should discussed the
results with articles of other research groups and not their own. This could enrich the
discussion.
Thank you for your question. We have proposed some criteria that emerge after
carefully evaluation of previous studies in the hope they would be helpful for
researchers/colleagues that are investigating PAs effects. In no case we have
transmitted this information to health authorities, policy makers or research
organizations, because we absolutely agree with you that they should be
validated or discussed by the scientific community

Round 2
Reviewer 3 Report
Regarding the manuscript, Proanthocyanidins: Intestinal Action Mechanisms in the Prevention and Treatment of Metabolic Syndrome.
I have now looked at the responses of authors to my comments. I feel that the authors have done their best to answer my concerns.
I would like to give me decision as Accept in its present form.